# Provable Overlapping Community Detection in Weighted Graphs

**Jimit Majmudar**
University of Waterloo
jmajmuda@uwaterloo.ca

**Stephen Vavasis**
University of Waterloo
vavasis@uwaterloo.ca

## Abstract

Community detection is a widely-studied unsupervised learning problem in which the task is to group similar entities together based on observed pairwise entity interactions. This problem has applications in diverse domains such as social network analysis and computational biology. There is a significant amount of literature studying this problem under the assumption that the communities do not overlap. When the communities are allowed to overlap, often a *pure nodes* assumption is made, i.e. each community has a node that belongs exclusively to that community. This assumption, however, may not always be satisfied in practice. In this paper, we provide a provable method to detect overlapping communities in weighted graphs without explicitly making the pure nodes assumption. Moreover, contrary to most existing algorithms, our approach is based on convex optimization, for which many useful theoretical properties are already known. We demonstrate the success of our algorithm on artificial and real-world datasets.

## 1 Introduction

Given a graph, determining subsets of vertices that are closely related in some sense is a problem of interest to many researchers. The two most common titles, in unsupervised learning, for problems of such variety are "community detection in graphs" and "graph clustering". These problems, due to their fundamental nature, arise in more than one domains. Some examples are: determining social circles in a social network (Du et al. [2007], Mishra et al. [2007], Bedi and Sharma [2016]), identifying functional modules in biological networks such as protein-protein interaction networks (Nepusz et al. [2012]), and finding groups of webpages on the World Wide Web that have content on similar topics (Dourisboure et al. [2009]).

The *Stochastic Block Model (SBM)* is a common mathematical framework for community detection, an in-depth survey of which can be found in Abbe [2017]. The SBM literature has a vast number of recovery guarantees such as those by Rohe et al. [2011], Lei et al. [2015], Li et al. [2018]. However, an obvious shortcoming of SBM is that it allows the nodes to belong to exactly one community. In practice, such an assumption is rarely satisfied. For example, in social network analysis, it is expected that some agents belong to multiple social circles or interest groups. Similarly, in the problem of clustering webpages, it is plausible that some webpages span multiple topics. Airoldi et al. [2008] proposed an extension of SBM, called the *Mixed Membership Stochastic Blockmodel (MMSB)*, in which nodes are allowed to have memberships in multiple communities. MMSB generalizes the traditional SBM by positing that each node may have fractional memberships in the different communities. If $n$ and $k$ denote the number of nodes and the number of communities respectively, matrix $\Theta \in [0,1]^{n \times k}$, called the *node-community distribution matrix*, is generated such that each of its rows is drawn from the Dirichlet distribution with parameters $\boldsymbol{\alpha} \in \mathbb{R}^k$. Then the $n \times n$ *probability matrix* is given as

$$P = \Theta B \Theta^T \tag{1}$$

where $B$ is a $k \times k$ *community interaction matrix*. Lastly, a random graph according to MMSB is generated on $n$ nodes by placing an edge between nodes $i$ and $j$ with probability $P_{ij}$. MMSB has been shown to be effective in many real-world settings, but the recovery guarantees regarding it are very limited compared to the SBM.

For a theoretical analysis of the MMSB, it is usually assumed that the user has access to only an unweighted random graph generated according to the model. While this assumption may be necessary in some settings, it makes the analysis difficult without much advantage. Indeed in many settings of practical interest, the user does have access to a similarity measure between node pairs, and this motivates us to work with weighted graphs generated according to the MMSB. For example, in the context of social network analysis, one may define a *communication graph* as an unweighted graph in which edge $ij$ exists if and only if agents $i$ and $j$ exchanged messages in a certain fixed time window. Then the weighted adjacency matrix for the social network may be obtained by averaging the adjacency matrices of multiple observed communication graphs. On the other hand, we make the problem difficult in a more realistic manner; we remove a common assumption in literature which is quite unrealistic if not mathematically problematic. This assumption requires each community in the input graph to contain a node which belongs exclusively to that community. Such nodes are called *pure nodes* in the literature. The notion of pure nodes in community detection is related to that of *separability* in nonnegative matrix factorization in the sense that they both induce a simplicial structure on the data. Although we do not make the pure nodes assumption, the Dirichlet distribution naturally generates increasingly better approximations to pure nodes as $n$, the number of nodes, gets large, and we use this fact in our analysis. As far as we know, this exact setup has not been studied before.

**Our Contributions:** We provide a simple provable algorithm for the recovery of $\Theta$ in the MMSB without explicitly requiring the communities to have pure nodes. Moreover, unlike most existing methods, our algorithm enjoys the merit of being rooted in linear programming. Indeed, multiple convex relaxations exist for SBM, but that is not the case for MMSB. As a byproduct of our analysis, we provide concentration results for some key random variables associated with the MMSB. We also demonstrate the applicability of MMSB, and consequently of our algorithm, to a problem of significant consequence in computational biology which is that of *protein complex detection* via experimental results using real-world datasets.

**Existing Provable Methods:** Zhang et al. [2014] propose the so-called *Overlapping Communities Community Assignment Model (OCCAM)* which only slightly differs from MMSB; in OCCAM each row of $\Theta$ has unit $\ell_2$-norm as opposed to unit $\ell_1$-norm in MMSB. They provide a provable algorithm for learning the OCCAM parameters in which one performs $k$-medians clustering on the rows of the $n \times k$ matrix corresponding to $k$ largest eigenvectors of the adjacency matrix corresponding to the observed unweighted random graph. However, their assumptions may be difficult to verify in practice. Indeed for their $k$-medians clustering to succeed, they assume that the ground-truth community structure provides the unique global optimal solution of their chosen $k$-medians loss function, which is also required to satisfy a special curvature condition around this minimum.

A moment-based tensor spectral approach to recover the MMSB parameters $\Theta$ and $B$ from an unweighted random graph generated according to the model was shown by Anandkumar et al. [2014]. Their approach, however, is not very straightforward to implement and involves multiple tuning parameters. Indeed one of the tuning parameters must be close to the sum of the $k$ Dirichlet parameters, which are not known in advance.

In a series of works, Mao et al. [2017a,b, 2018] have also tackled the problem of learning the parameters in MMSB from a single random graph generated by the model. However, they require the pure node assumption. Additionally, they cast the MMSB recovery problem as problems that are nonconvex. Consequently, to get around the nonconvexity, more assumptions on the model parameters are required. For instance, in Mao et al. [2017a], the MMSB recovery problem is formulated as a symmetric nonnegative matrix factorization (SNMF) problem, which is both nonconvex and NP-hard. Then to ensure the uniqueness of the global optimal solution for the SNMF problem, they require $B$ to be a diagonal matrix. In contrast, not only does our approach directly tackle the factorization in (1) to recover $\Theta$, we also do so using linear programming.

Recently Huang and Fu [2019] have also proposed a linear programming-based algorithm for recovery in MMSB. However, the connection between their proposed linear programs and ours is unclear, and they require the pure nodes assumption for their method to provably recover the communities.

**Notation**: For any matrix $M$, we use $\mathbf{m}_i$ and $\mathbf{m}^i$ to denote its column $i$ and the transpose of its row $i$ respectively, and $M_{ij}$ to denote its entry $ij$; for any set $\mathcal{R} \subseteq [n]$, $M(\mathcal{R}, :)$ denotes the submatrix of $M$ containing all columns but only the rows indexed by $\mathcal{R}$. We use $\| \cdot \|$ to denote the $\ell_2$-norm for vectors and the spectral norm (largest singular value) for matrices. For a matrix, $\| \cdot \|_{\max}$ denotes its largest absolute value. $I$ denotes the identity matrix whose dimension will be clear from the context. For any positive integer $i$, $\mathbf{e}_i$ denotes column $i$ of the identity matrix and $\mathbf{e}$ denotes the vector with each entry set to one; the dimension of these vectors will be clear from context.

## 2 Proposed Work

### 2.1 Problem Formulation

We ask the following question for the MMSB described by (1):

*Given $P$, how can we efficiently obtain a matrix $\hat{\Theta} \in [0, 1]^{n \times k}$ such that $\hat{\Theta} \approx \Theta$?*

Typically, one imposes the pure nodes assumption on $\Theta$ which greatly simplifies the above posed problem. That is, one assumes that for each $j \in [k]$, there exists $i \in [n]$ such that $\boldsymbol{\theta}^i = \mathbf{e}_j$, i.e. node $i$ belongs exclusively to community $j$. In other words, the rows of $\Theta$ contain all corners of unit simplex in $\mathbb{R}^k$. However, such an assumption is mathematically problematic and/or practically unrealistic. Indeed if the rows of $\Theta$ are sampled from the Dirichlet distribution, then the probability of sampling even one pure node is zero. Moreover, even from a practical standpoint such an assumption may not always be satisfied since in real-world networks, such as protein-protein interaction networks, one encounters communities with no pure nodes. Lastly, note that we are interested in recovering only $\Theta$ and not $B$ since the former contains the community membership information of each node which is usually what a user of such methods is interested in.

We provide an answer to posed question without making an explicit assumption regarding the presence of pure nodes. To that effect, we propose a novel simple and efficient convex optimization-based method to approximate $\Theta$ entrywise under a very natural condition that just requires $n$ to be sufficiently large. Such a condition is often satisfied in practice since real-world graphs in application settings such as social network analysis are usually large-scale.

**Identifiability:** It is known that having pure nodes for each community is both necessary and sufficient for identifiability of MMSB (Mao et al. [2020]). Since we make no assumption regarding the presence of pure nodes in this work, we cannot necessarily expect MMSB to be identifable. However, as a result of our analysis, we are able to show that for sufficiently large graphs, if there exist two distinct sets of parameters for the MMSB which yield the same probability matrix $P$, then their corresponding node-community distribution matrices are sufficiently close to each other with high probability. This notion, which is formalized in Corollary 3.2, may be interpreted as *near identifiability*.

### 2.2 SP+LP **Recovery Algorithm**

We may think of our recovery procedure, *Successive Projection followed by Linear Programming* (SP+LP), as divided into two stages. First, via a preprocessing step, called Successive Projection, we obtain a set $\mathcal{J} \subseteq [n]$ of cardinality $k$ such that $\Theta(\mathcal{J}, :)$ is entrywise close to $I$ up to a permutation of the rows. We may think of the nodes in $\mathcal{J}$ as being *almost pure*, which we then we use to recover approximations to the $k$ columns of $\Theta$, the *community characteristic vectors*, using exactly $k$ linear programs (LPs). The form of the LP in SP+LP can be motivated as follows. Intuitively, the presence of almost pure nodes ensures that the column range of $\Theta$ coincides with the range of $P$; consequently recovering $\Theta$ given $P$ may be interpreted as obtaining a certain basis for the range of $P$. These desired basis vectors, i.e. the columns of $\Theta$, are nonnegative and somewhat sparse in the sense that they contain potentially many entries which are close to zero. Thus we seek nonnegative vectors in the range of $P$ with the smallest $\ell_1$-norm (which, for a nonnegative vector, is equal to the sum of its entries) and introduce a non-homogeneous constraint to rule out the trivial solution of the zero vector. Similar optimization formulation techniques have been shown to work for some planted/generative models for the problem for sparse dictionary learning (Spielman et al. [2012], Qu et al. [2014]). Note that SP+LP has no tuning parameters other than the number of communities, which is also a parameter for most other community detection algorithms.

**Algorithm 1** SP+LP
***

**Input:** Matrix $P$ generated according to MMSB, number of communities $k$
**Output:** Estimated characteristic vectors $\hat{\boldsymbol{\theta}}_1, \ldots, \hat{\boldsymbol{\theta}}_k \in [0,1]^n$

  1:  $\mathcal{J} = \text{SuccessiveProjection}(P)$
  2: **for** $i \in [k]$ **do**
  3:    $(\mathbf{x}^*, \mathbf{y}^*) = \arg\min\limits_{(\mathbf{x}, \mathbf{y})} \mathbf{e}^T \mathbf{x}$  s.t.  $\mathbf{x} \geq \mathbf{0}, x_{\mathcal{J}(i)} \geq 1, \mathbf{x} = P\mathbf{y}$
  4:    $\hat{\boldsymbol{\theta}}_i = \mathbf{x}^* / \|\mathbf{x}^*\|_\infty$
  5: **end for**
***

**Algorithm 2** SuccessiveProjection
***

**Input:** Matrix $P$ generated according to MMSB, number of communities $k$
**Output:** Estimated set of almost pure nodes $\mathcal{J} \subseteq [n]$

  1:  $\mathcal{J} = \{\}, R = P, j = 1$
  2: **while** $R \neq 0$ and $j \in [k]$ **do**
  3:    $s' = \arg\max\limits_{s \in [n]} \|\mathbf{p}_s\|^2$
  4:
  5:    $R = \left( I - \dfrac{\mathbf{p}_{s'} \mathbf{p}_{s'}^T}{\|\mathbf{p}_{s'}\|^2} \right) R$
  6:    $\mathcal{J} = \mathcal{J} \cup \{s'\}$
  7:    $j = j + 1$
  8: **end while**
***

## 3 Theoretical Guarantees

Let $Z$ be a $k \times k$ submatrix of $\Theta$ such that for each $j \in [k]$, there exists $i \in [k]$ satisfying

$$\|\mathbf{z}^i - \mathbf{e}_j\|_\infty \leq \|\boldsymbol{\theta}^p - \mathbf{e}_j\|_\infty \tag{2}$$

for any $p \in [n]$. The rows of $Z$ do not exactly correspond to the corners of the unit simplex; they are, however, the best entrywise approximations of the corners that can be obtained among the rows of $\Theta$. Note that without loss of generality, through appropriate relabelling of the nodes, we may assume that indices $i$ and $j$ in (2) are identical. Define the $k \times k$ matrix $\Delta := Z - I$.

Define $\mathbf{c} := \Theta^T \mathbf{e}$, and let $c_{\min}$ and $c_{\max}$ denote the smallest and largest entries in $\mathbf{c}$ respectively. Let $\kappa$ and $\kappa_0$ denote the condition numbers of $B$ and $\Theta B$ respectively, associated with the $\ell_2$-norm.

Now we state our main result, which provides complete theoretical justification for the success of SP+LP in approximately recovering the $k$ community vectors.

**Theorem 3.1.** *Suppose $k \geq 2$, $B$ is full-rank, and all $k$ parameters of the Dirichlet distribution are equal to $\alpha \in \mathbb{R}$. Let $w := 8\kappa\sqrt{\alpha k + 1}$ and define*

$$\epsilon_1 := \min\left( \frac{1}{\sqrt{k-1}}, \frac{1}{2} \right) \frac{1}{2\sqrt{2}w(1 + 80w^2)}$$

$$\epsilon_2 := \frac{7}{3520\sqrt{2}kw^2}.$$

*If $n > \dfrac{\log(p/k)}{\log I_{1-\epsilon}(\alpha, (k-1)\alpha)}$ for some $p \in (0,1)$ and $\epsilon \in (0, \min\{\epsilon_1, \epsilon_2\})$, then there exists a permutation $\pi$ of the set $[k]$ such that vectors $\hat{\boldsymbol{\theta}}_1, \ldots, \hat{\boldsymbol{\theta}}_k$ returned by SP+LP satisfy*

$$\max_{j \in [k]} \|\hat{\boldsymbol{\theta}}_j - \boldsymbol{\theta}_{\pi(j)}\|_\infty = \mathcal{O}(\alpha k^2 \kappa^2 \epsilon) \tag{3}$$

*with probability at least $1 - p - c_1 e^{-c_2 n}$ where $c_1, c_2$ are constants that depend on $\alpha, k, \kappa$.*

*(Here $I_x(y, z)$ denotes the regularized incomplete beta function.)*

We note that even though our main result is stated for an equal parameter Dirichlet distribution, our proof techniques simply extend, in principle, to a setting in which the Dirichlet parameters are

different but not too far from each other. Doing so, however, adds only incremental value but makes the analysis significantly tedious.

Before presenting further results about our recovery procedure, we present a result regarding the near identifiability of the MMSB without the presence of pure nodes for each community.

**Corollary 3.2.** *Let $(n, k, \alpha, B)$ and $(n, k, \bar{\alpha}, \bar{B})$ be two distinct sets of parameters for the MMSB such that $\kappa$ and $\bar{\kappa}$ denote the condition numbers of $B$ and $\bar{B}$ respectively, and $\epsilon_1, \epsilon_2$ and $\bar{\epsilon}_1, \bar{\epsilon}_2$ are defined respectively for the two sets as in Theorem 3.1. Moreover, suppose that these two sets satisfy the conditions of Theorem 3.1 for some $p, \epsilon$ and $\bar{p}, \bar{\epsilon}$ such that $p, \bar{p} \in (0, 1)$, $\epsilon \in (0, \min\{\epsilon_1, \epsilon_2\})$ and $\bar{\epsilon} \in (0, \min\{\bar{\epsilon}_1, \bar{\epsilon}_2\})$. If the respective node-community distribution matrices are $\Theta$ and $\bar{\Theta}$ such that $\Theta B \Theta^T = \bar{\Theta} \bar{B} \bar{\Theta}^T$, then there exists a permutation $\pi$ of the set $[k]$ such that*

$$\max_{j \in [k]} \|\bar{\boldsymbol{\theta}}_j - \boldsymbol{\theta}_{\pi(j)}\|_\infty = \mathcal{O}(k^2(\alpha \kappa^2 \epsilon + \bar{\alpha} \bar{\kappa}^2 \bar{\epsilon})) \tag{4}$$

*with probability at least $1 - p - \bar{p} - c_1 e^{-c_2 n} - \bar{c}_1 e^{-\bar{c}_2 n}$ where $c_1, c_2$ and $\bar{c}_1, \bar{c}_2$ are constants that depend on $\alpha, k, \kappa$ and $\bar{\alpha}, k, \bar{\kappa}$ respectively.*

In line with our algorithm description, we divide the theoretical analysis also in two parts: one for analysis of the preprocessing Successive Projection subroutine, and another for analysis of the LPs in the main algorithm.

Successive Projection Algorithm was first studied by Gillis and Vavasis [2013] in far more generality than what is used here. Adopting their main recovery theorem to our setup yields the following theorem.

**Theorem 3.3** (Gillis and Vavasis [2013]). *Suppose that*

$$\|\Delta\|_{\max} < \min\left(\frac{1}{\sqrt{k-1}}, \frac{1}{2}\right) \frac{1}{2\sqrt{2}\kappa_0(1 + 80\kappa_0^2)} \tag{5}$$

*and let $\mathcal{J}$ be the index set of cardinality $k$ extracted by Algorithm 2. Then there exists a $k \times k$ permutation matrix $\Pi$ such that*

$$\|\Pi\Theta(\mathcal{J}, :) - I\|_{\max} \leq 40\sqrt{2}\kappa_0^2\|\Delta\|_{\max}. \tag{6}$$

Theorem 3.3 provides theoretical justification for the success of the subroutine highlighted in Algorithm 2. To this end, our contribution is to show that the condition in (5) is satisfied in MMSB with high probability provided the number of nodes in the graph is sufficiently large. This involves deriving concentration bounds for the smallest and largest singular values of $\Theta$ and $\Theta B$. The following result provides theoretical guarantee for the performance of the LP in Algorithm 1.

**Theorem 3.4.** *Assume $k \geq 2$, $B$ is full-rank, and $c_{\min}/c_{\max} > 1/2$. Suppose for each $s \in [k]$, there exists $p \in [n]$ such that $\|\boldsymbol{\theta}^p - \mathbf{e}_s\|_\infty \leq \eta$ for some $0 \leq \eta < (c_{\min}/c_{\max} - 1/2)/4k$.*

*Let $i \in [n]$ such that $\|\boldsymbol{\theta}^i - \mathbf{e}_j\|_\infty \leq \eta$ for some $j \in [k]$. Then the LP*

$$\begin{aligned}
\min \quad & \mathbf{e}^T \mathbf{x} \\
\text{s.t.} \quad & \mathbf{x} \geq \mathbf{0} \\
& x_i \geq 1 \\
& \mathbf{x} = P\mathbf{y}
\end{aligned} \tag{P}$$

*has an optimal solution, and if $\mathbf{x}^*$ is an optimal solution then*

$$\left\|\frac{\mathbf{x}^*}{\|\mathbf{x}^*\|_\infty} - \boldsymbol{\theta}_j\right\|_\infty \leq 4\eta(2\sqrt{2}k + 1). \tag{7}$$

*Moreover, the time complexity of solving (P) to obtain $x^*$ is $\mathcal{O}(n^2)$.*

Combining Theorems 3.3 and 3.4 yields Theorem 3.1, which provides entrywise error bounds for the $k$ community characteristic vectors returned by SP+LP. We also conclude that the time complexity of SP+LP is $\mathcal{O}(n^2)$ since the time complexity of both SuccessiveProjection and solving (P) is $\mathcal{O}(n^2)$. To our best knowledge, there does not exist a competing provable algorithm whose time complexity is under $\mathcal{O}(n^2)$.

# 4 Experiments

In this section, we compare the performance of SP+LP on both synthetic and real-world graphs with other popular algorithms. In practice, the user has access to the adjacency matrix, called $A$, of the observed weighted graph which is only an approximation of $P$. Matrix $A$ may even be full-rank, and so for implementation we have to slightly modify the constraint $\mathbf{x} = P\mathbf{y}$ in the LP in SP+LP. (Indeed note that if $A$ is full-rank, then the optimal solution to the LP is $\mathbf{e}_{\mathcal{J}(i)}$.) Specifically, we replace that constraint with $\mathbf{x} = V\mathbf{y}$ where $V$ is an $n \times k$ matrix whose columns contain the eigenvectors of $A$ corresponding to either its $k$ largest eigenvalues or singular values. The intuition behind this is that we expect the range of $V$ to approximate the $k$-dimensional subspace of $\mathbb{R}^n$ which is the range of $P$. For efficient computation of $V$, one may employ, for instance, the Lanczos method or randomized SVD (Halko et al. [2011]).

## 4.1 Synthetic Graphs

We demonstrate the performance of SP+LP on artificial graphs generated according to the MMSB. In practice, the weighted adjacency matrix available is only approximately equal to $P$. Therefore for our experiments, we compute a weighted adjacency matrix by averaging $s$ number of $0, 1$-adjacency matrices, each of which is sampled according to $P$. That is, entry $ij$ of a sampled adjacency matrix is a Bernoulli random variable with parameter $P_{ij}$. The diagonal entries in these adjacency matrices are all set to $1$.

**Evaluation Metrics:** We evaluate SP+LP in terms of the entrywise error in the predicted columns of $\Theta$ and the wall-clock running time (Figure 1). The entrywise error is defined as $\min_{\Pi} \|\hat{\Theta} - \Theta\Pi\|_{\max}$ over all $k \times k$ permutation matrices $\Pi$, where $\hat{\Theta} := \begin{bmatrix} \hat{\boldsymbol{\theta}}_1 & \dots & \hat{\boldsymbol{\theta}}_k \end{bmatrix}$ contains as columns the predicted community characteristic vectors. For each plot, each point is determined by averaging the results over 10 samples and the error bars represent one standard deviation.

We compare our results with the GeoNMF algorithm which has been shown in Mao et al. [2017a] to computationally outperform popular methods such as Stochastic Variational Inference (SVI) by Gopalan and Blei [2013], a Bayesian variant of SNMF by Psorakis et al. [2011], the OCCAM algorithm by Zhang et al. [2014], and the SAAC algorithm by Kaufmann et al. [2016]. We use the implementation of GeoNMF that is made available by the authors without any modification and also the provided default values for the tuning parameters.

**Parameter Settings:** Unless otherwise stated, the default parameter settings are $n = 5000, k = 3, \alpha = 0.5, s = \sqrt{n}$. Figures 1(d) and 1(f) show the performance of the SP+LP for community interaction matrices $B$ with higher off-diagonal elements. More specifically, for those plots, we set $B = (1 - \delta) \cdot I + \delta \cdot \mathbf{e}\mathbf{e}^T$. For Figures 1(a), 1(b), 1(c), 1(e), we set $B = 0.5 \cdot I + 0.5 \cdot R$ where $R$ is a $k \times k$ diagonal matrix whose each diagonal entry is generated from a uniform distribution over $[0, 1]$. One reason for choosing these parameter settings is to have a fair comparison. Indeed GeoNMF has already been shown to perform well over these parameter choices.

Figures 1(a), 1(b), 1(c), 1(d) demonstrate that SP+LP outperforms GeoNMF in terms of the entrywise error in the recovered MMSB communities with increasing $n, k, \alpha$ and $\delta$. In particular, this implies that, compared to GeoNMF, SP+LP can handle larger graphs, more number of communities, more overlap among the communities, and a more general community interaction matrix $B$, while involving a lesser number of tuning parameters. However, Figure 1(e) shows that SP+LP is slower compared to GeoNMF and that opens up possibilities for future work to expedite SP+LP. On the other hand, Figure 1(f) shows that for a more general $B$, the time performances of GeoNMF and SP+LP are quite comparable.

## 4.2 Real-world Graphs

For practical application of SP+LP, we consider a well-studied problem in computational biology: that of clustering functionally similar proteins together based on protein-protein interaction (PPI) observations (see Nepusz et al. [2012]). In the language of our problem setup, each node in the weighted graph represents a protein, and the weights represent the reliability with which any two proteins interact. The communities or clusters of similar proteins are called *protein complexes* in biology literature.

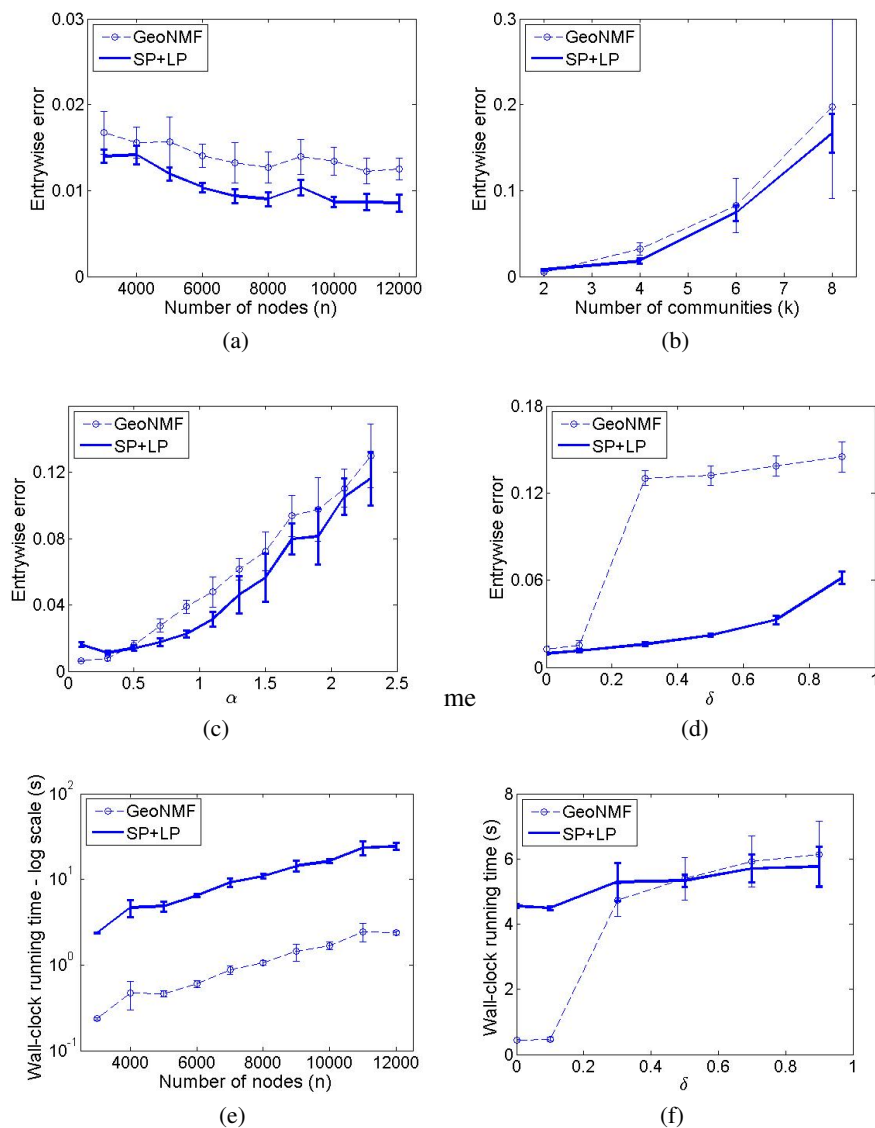

me

Figure 1: Performance of SP+LP on synthetic MMSB weighted graphs compared with GeoNMF.

It is important to highlight that the PPI networks typically contain a large number of communities compared to the number of nodes and therefore our theory does not necessarily guarantee that SP+LP will succeed with high probability. Despite that, we observe that on some datasets, SP+LP matches or even outperforms commonly-used protein complex detection heuristics. Additionally, protein complex detection is a very well-studied problem in biology and there exist a vast number of heuristics which are tailored for this specific problem. For instance, recent works of Yu et al. [2013], Yu et al. [2014], and Yu et al. [2015] incorporate existing ground truth knowledge of protein complexes in the algorithm to obtain a supervised learning-based approach. Our goal in this paper is not to design a fine-tuned method specifically for protein complex detection. We are focused on studying the general purpose MMSB with minimal assumptions and demonstrating its applicability to a real-world problem of immense consequence. The connection of MMSB with protein complex detection was also made in Airoldi et al. [2006]; however, their theoretical and experimental results are quite preliminary compared to ours.

**Datasets:** We consider PPI datasets provided by Krogan et al. [2006] and Collins et al. [2007], which are very popular among the biological community for the protein complex detection problem. The

Table 1: Comparision of SP+LP with ClusterONE on Krogan core, Krogan extended, and Gavin datasets using SGD repository as validation set.

| Validation set | Metric | Krogan core | | Krogan extended | | Collins | |
|---|---|---|---|---|---|---|---|
| | | SP+LP | ClusterONE | SP+LP | ClusterONE | SP+LP | ClusterONE |
| SGD | MMR | 0.389 | 0.418 | 0.428 | 0.364 | 0.372 | 0.532 |
| | frac | 0.598 | 0.667 | 0.632 | 0.594 | 0.557 | 0.828 |
| | GA | 0.525 | 0.663 | 0.542 | 0.628 | 0.504 | 0.731 |
| | Score | 1.512 | 1.748 | 1.602 | 1.586 | 1.433 | 2.091 |

Table 2: Comparision of SP+LP with ClusterONE on Krogan core, Krogan extended, and Gavin datasets using MIPS repository as validation set.

| Validation set | Metric | Krogan core | | Krogan extended | | Collins | |
|---|---|---|---|---|---|---|---|
| | | SP+LP | ClusterONE | SP+LP | ClusterONE | SP+LP | ClusterONE |
| MIPS | MMR | 0.285 | 0.317 | 0.319 | 0.282 | 0.275 | 0.418 |
| | frac | 0.537 | 0.669 | 0.576 | 0.573 | 0.547 | 0.782 |
| | GA | 0.331 | 0.438 | 0.336 | 0.422 | 0.397 | 0.555 |
| | Score | 1.153 | 1.424 | 1.231 | 1.277 | 1.219 | 1.755 |

former contains two weighted graph datasets, which are referred to as Krogan core ($n = 2708$) and Krogan extended ($n = 3672$). The weighted graph dataset in the latter is referred to as Collins ($n = 1622$). The ground truth validation sets used are two standard repositories of protein complexes, which also appear to be the benchmarks in the biological community. These repositories are Munich Information Centre for Protein Sequence (MIPS) and *Saccharomyces* Genome Database (SGD). These repositories are manually curated and therefore are independent of the PPI datasets. We emphasize that protein complex detection is an ongoing research effort and that these repositories may not necessarily be considered complete as yet. This implies that SP+LP may find candidate complexes that are not known thus far but nonetheless do exist, thereby acting as a tool for biologists to make educated guesses.

**Evaluation Metrics:** The success of a protein complex detection algorithm is typically measured via a composite score which is the sum of three quantities: maximum matching ratio (MMR), fraction of detected complexes (frac), and geometric accuracy (GA). Intuitively, MMR captures how well the complexes in the validation set are predicted by computing a maximum matching in a bipartite graph in which the two vertex sets represent predicted and true complexes, and the weight of an edge denotes a similarity score between the predicted complex and the true complex on its endpoints, frac captures the fraction of true complexes for which a sufficiently good predicted complex exists, and GA is the geometric mean of clustering-wise sensitivity and positive predictive value. The reader may refer Nepusz et al. [2012] for an excellent in-depth discussion about these quantities. A higher score corresponds to better performance and the highest possible scores for MMR and frac are one each.

For the parameter $k$, we try different plausible values. The validation sets have binary memberships for the protein complexes, i.e. each protein is either present in a complex or it is not. The memberships determined via SP+LP, on the other hand, are fractional. However, the former can be easily binarized by rounding all entries that are at least $0.5$ to 1 and rounding the remaining entries to 0. Additionally, we have performed another post-processing step after binarizing the result of SP+LP which appears quite commonly in the domain literature. Any pair of complexes that overlap significantly (as determined by a user-defined threshold) are merged. Tables 1 and 2 show the performance of SP+LP for protein complex detection, and we compare our results with one of the most popular problem-specific heuristics called ClusterONE. We highlight that, unlike MMR and frac, GA penalizes predictions which contain complexes in addition to the true complexes. Therefore such a metric is not suitable for any algorithm which might predict new potentially valid complexes that do not yet exist in the validation sets.

# 5 Conclusions

In this work, we show how to detect potentially overlapping communities in a setup that is more plausible in real-world applications, i.e. in weighted graphs without assuming the presence of pure nodes. Our method uses linear programming, which is a relatively principled approach since the literature on the theory of convex optimization is quite rich. We show that our method performs excellently on synthetic datasets. Additionally, we also show that our method succeeds in solving an important problem in computational biology without any major domain-specific modifications to the algorithm.

# 6 Broader Impact

The problem of community detection (or graph clustering) is a standard abstraction that has been studied by various research communities for a few decades now. This work presents a provable overlapping community detection method which may possibly be used, for instance, by biologists to discover new potentially overlapping protein complexes. Moreover, the proposed work is fairly general and does not present any foreseeable negative societal consequences.

## Acknowledgments and Disclosure of Funding

Support for this work was provided by a Discovery Grant from the Natural Sciences and Engineering Research Council (NSERC) of Canada and by internal University of Waterloo funds. JM held a short-term paid position collaborating with researchers at Google investigating another area of AI before this work began.

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
