[Supplementary Material · Supplementary - Provable Overlapping Community Detection.pdf]

# Supplementary Material for "Provable Overlapping Community Detection in Weighted Graphs"

## A   LP Analysis

Let $\eta \in (0, 1)$ and assume for now that for each $j \in [k]$, there exists $i \in [n]$ such that $\|\boldsymbol{\theta}^i - \mathbf{e}_j\|_\infty \leq \eta$. Moreover, assume, without loss of generality, that for each $i \in [k]$

$$\|\boldsymbol{\theta}^i - \mathbf{e}_i\|_\infty \leq \eta. \tag{8}$$

Indeed such a property can always be satisfied with appropriate relabelling of the nodes. Define $I' := \Theta([k], :)$.

**Lemma A.1.** *Suppose $M$ is a $k \times k$ matrix whose rows belong to the unit simplex. If*

$$\|\mathbf{m}^i - \mathbf{e}_i\|_\infty \leq \delta \tag{9}$$

*for each $i \in [k]$ and for some $\delta \in \left[0, \dfrac{1}{2\sqrt{2k}}\right]$, then*

$$\|M^{-T} - I\|_\infty \leq 2\sqrt{2}\delta k. \tag{10}$$

*Proof.* Since each row of $M$ belongs to the unit simplex and satisfies (9), we note that $\ell_2$-norm of each column of $M - I$ is bounded above by $\delta\sqrt{2}$. This implies that

$$\|M - I\| \leq \delta\sqrt{2k}. \tag{11}$$

Moreover

$$
\begin{aligned}
\left|\|M^{-1}\| - 1\right| &\leq \|M^{-1} - I\| && \text{(using reverse triangle inequality)} \\
&= \|(M - I)M^{-1}\| \\
&\leq \|M - I\|\|M^{-1}\|
\end{aligned}
$$

which implies that

$$\|M^{-1}\| \leq \frac{1}{1 - \|M - I\|}. \tag{12}$$

Then, we have

$$
\begin{aligned}
\|M^{-T} - I\|_\infty &\leq \sqrt{k}\|M^{-T} - I\| \\
&= \sqrt{k}\|M^{-1} - I\| \\
&\leq \sqrt{k}\|M - I\|\|M^{-1}\| \\
&\leq \frac{\sqrt{k}\|M - I\|}{1 - \|M - I\|} && \text{(using (12))} \\
&\leq \frac{\sqrt{2}\delta k}{1 - \delta\sqrt{2k}} \\
&\leq 2\sqrt{2}\delta k. && \text{(by assumption on $\delta$)}
\end{aligned}
$$

$\square$

For any $i \in [k]$, consider the LP

$$\begin{aligned} \min \quad & \mathbf{c}^T \mathbf{y} \\ \text{s.t.} \quad & \Theta \mathbf{y} \geq \mathbf{0} \\ & \mathbf{y}^T \boldsymbol{\theta}^i \geq 1. \end{aligned} \tag{Pi}$$

and its dual

$$\begin{aligned} \max \quad & \beta \\ \text{s.t.} \quad & \beta \boldsymbol{\theta}^i + \Theta^T \mathbf{u} = \mathbf{c} \\ & \beta, \mathbf{u} \geq 0. \end{aligned} \tag{Di}$$

Note that both (Pi) and (Di) are feasible optimization problems. Thus let $\mathbf{y}^*$ and $(\beta^*, \mathbf{u}^*)$ be a (Pi)-(Di) optimal solution pair.

**Lemma A.2.** *Suppose* $\eta \leq \dfrac{1}{2\sqrt{2}k} \dfrac{c_{\min}}{c_{\max}}$.

*Then*

$$c_i - 2\sqrt{2}\eta k c_{\max} \leq \beta^* \leq \frac{c_i}{1 - \eta}. \tag{13}$$

*Proof.* The upper bound follows from observing that $\beta^* = \mathbf{c}^T \mathbf{y}^*$ due to Strong Duality and that $\mathbf{e}_i/\theta_{ii}$ is a feasible solution for (Pi), combined with the fact that $\theta_{ii} \geq 1 - \eta$.

For the lower bound we construct a feasible solution for (Di). Define $\mathbf{z}$ as the solution of the system $I'^T \mathbf{z} = \mathbf{c}$. Note that the rows of $I'$ belong to the unit simplex and for any $i \in [k]$, we have

$$\begin{aligned} \|I'(i, :) - \mathbf{e}^i\|_\infty &\leq \eta \\ &\leq \frac{1}{2\sqrt{2}k}. \qquad \text{(by assumption on } \eta) \end{aligned}$$

Therefore using Lemma A.1, we conclude that $\|I'^{-T} - I\|_\infty \leq 2\sqrt{2}\eta k$.

Then for any $s \in [k]$, we have

$$\begin{aligned} |z_s - c_s| &\leq \|\mathbf{z} - \mathbf{c}\|_\infty \\ &\leq \|I'^{-T} - I\|_\infty c_{\max} \\ &\leq 2\sqrt{2}\eta k c_{\max}. \end{aligned}$$

Moreover since $\eta \leq \dfrac{1}{2\sqrt{2}k} \dfrac{c_{\min}}{c_{\max}}$, we conclude that $\mathbf{z} \geq \mathbf{0}$. Now define the point $(\beta', \mathbf{u}')$ such that

$$\beta' := z_i$$

and

$$u'_s := \begin{cases} z_s, & \text{if } s \in [k] \setminus \{i\} \\ 0, & \text{otherwise.} \end{cases}$$

Note that $(\beta', \mathbf{u}')$ is feasible for (Di) with objective value

$$\beta' \geq c_i - 2\sqrt{2}\eta k c_{\max}.$$

$\square$

Define the vector $\mathbf{r} := \Theta^T \mathbf{u}^*/2$. We shall prove some bounds on the entries of $\mathbf{r}$ which will be used for subsequent proofs.

**Lemma A.3.** *Suppose* $\eta \leq \dfrac{1}{2\sqrt{2}k} \dfrac{c_{\min}}{c_{\max}}$. *Then we have the following inequalities.*

1. $0 \leq r_i \leq 2k\eta c_{\max}$.

2. *For any* $s \in [k] \setminus \{i\}$,

$$c_{\min} - \frac{\eta}{1 - \eta} c_{\max} \leq r_s \leq \frac{c_{\max}}{2}.$$

*Proof.* First note that $\mathbf{r} \geq \mathbf{0}$ by definition and therefore the lower bound on $r_i$ follows. From the feasibility of $(\beta^*, \mathbf{u}^*)$ for (Di), we have for any $s \in [k]$

$$r_s = \frac{c_s - \beta^* \theta_{is}}{2}. \tag{14}$$

The upper bound on $r_i$ follows from (14), and using the lower bound on $\beta^*$ from Lemma A.2 and the fact that $\theta_{ii} \geq 1 - \eta$. Indeed, we have

$$
\begin{aligned}
r_i &= \frac{c_i - \beta^* \theta_{ii}}{2} \\
&\leq \frac{c_i - [(c_i - 2\sqrt{2}\eta k c_{\max})(1 - \eta)]}{2} \\
&= \frac{\eta c_i + 2\sqrt{2}\eta(1 - \eta)k c_{\max}}{2} \\
&\leq \frac{\eta c_{\max}[1 + 2\sqrt{2}(1 - \eta)k]}{2} \\
&\leq \eta c_{\max}\left(\frac{1 + 3k}{2}\right) \\
&\leq 2k\eta c_{\max}. && (\because k \geq 2)
\end{aligned}
$$

For any $s \in [k] \setminus \{i\}$, the upper bound on $r_s$ follows from (14), and noting that $\beta^*$ and $\theta_{is}$ are nonnegative and $c_s \leq c_{\max}$.

For any $s \in [k] \setminus \{i\}$, the lower bound on $r_s$ follows from (14), and using the upper bound on $\beta^*$ from Lemma A.2, the fact that $c_s \geq c_{\min}$ and the fact that $\theta_{is} \leq \eta$. $\square$

**Lemma A.4.** *Suppose* $\eta \leq \dfrac{1}{3k}\dfrac{c_{\min}}{c_{\max}}$. *Then* $\|\mathbf{r}\|_\infty \leq \dfrac{c_{\max}}{2}$.

*Proof.* We prove this statement by proving that $\|\mathbf{r}\|_\infty$ is attained at some index in $[k] \setminus \{i\}$. It suffices to show that $r_i \leq r_s$ for any $s \in [k] \setminus \{i\}$. Note that by assumption $\eta \leq \dfrac{1}{3k}\dfrac{c_{\min}}{c_{\max}} \leq \dfrac{1}{2\sqrt{2}}\dfrac{c_{\min}}{c_{\max}}$ and therefore the entries of $\mathbf{r}$ are bounded according to Lemma A.3.

We have

$$
\begin{aligned}
c_{\min} &\geq 2\eta c_{\max}\frac{3k}{2} && \text{(by assumption on } \eta) \\
&\geq 2\eta c_{\max}(k + 1) && (\because k \geq 2) \\
&= 2\eta c_{\max} + 2k\eta c_{\max} \\
&\geq \frac{\eta}{1 - \eta}c_{\max} + 2k\eta c_{\max} && (\because \eta \leq 1/2)
\end{aligned}
$$

which is equivalent to

$$2k\eta c_{\max} \leq c_{\min} - \frac{\eta}{1 - \eta}c_{\max}.$$

Therefore using Lemma A.3, we conclude that $r_i \leq r_s$ for any $s \in [k] \setminus \{i\}$. $\square$

**Lemma A.5.** *Suppose* $\dfrac{c_{\min}}{c_{\max}} > \dfrac{1}{2}$ *and* $\eta < \dfrac{1}{3k}\left(\dfrac{c_{\min}}{c_{\max}} - \dfrac{1}{2}\right)$. *Then for any* $s \in [k] \setminus \{i\}$, *if* $y_s^*$ *is positive, we have*

$$y_s^* < 2\sqrt{2}\eta k. \tag{15}$$

*Proof.* Pick any $s \in [k] \setminus \{i\}$ such that $y_s^* > 0$. Consider the auxiliary LP

$$
\begin{aligned}
\min \quad & \mathbf{c}^T \mathbf{y} \\
\text{s.t.} \quad & \Theta \mathbf{y} \geq \mathbf{0} \\
& \mathbf{y}^T \boldsymbol{\theta}^i \geq 1 \\
& y_s \geq 2\sqrt{2}\eta k
\end{aligned}
\tag{Pi-aux}
$$

and its dual

$$
\begin{aligned}
\max \quad & \beta + (2\sqrt{2}\eta k)\gamma \\
\text{s.t.} \quad & \beta\boldsymbol{\theta}^i + \gamma\mathbf{e_s} + \Theta^T\mathbf{u} = \mathbf{c} \\
& \beta, \gamma, \mathbf{u} \geq 0.
\end{aligned}
\tag{Di-aux}
$$

If we show that $\mathbf{y}^*$ is not an optimal solution to (Pi-aux), then we can conclude that $y_s^* < 2\sqrt{2}\eta k$. Therefore our goal is to show that the optimal value of (Pi-aux) is greater than $\mathbf{c}^T\mathbf{y}^*$. Equivalently, we may also show that the optimal value of (Di-aux) is greater than $\beta^*$. We do so by constructing a feasible solution for (Di-aux) at which the objective value is greater than $\beta^*$.

Now define $\bar{I}$ to be identical to $I'$ except the $s^{th}$ row which is set to be $\mathbf{e_s}^T$. Let $\mathbf{z}^*$ be the solution to the system

$$
\bar{I}^T\mathbf{z} = \mathbf{r}
\tag{16}
$$

where recall that $\mathbf{r} = \Theta^T\mathbf{u}^*/2$.

Note that the rows of $\bar{I}$ belong to the unit simplex and for any $i \in [k]$, we have

$$
\begin{aligned}
\|\bar{I}(i,:) - \mathbf{e}^i\|_\infty &\leq \eta \\
&\leq \frac{1}{2\sqrt{2k}}. \quad \text{(by assumption on } \eta)
\end{aligned}
$$

Therefore using Lemma A.1, we conclude that

$$
\|\bar{I}^{-T} - I\|_\infty \leq 2\sqrt{2}\eta k.
\tag{17}
$$

Define the point

$$
\begin{bmatrix} \bar{\beta} \\ \bar{\gamma} \\ \bar{\mathbf{u}} \end{bmatrix} := \begin{bmatrix} \beta^* \\ 0 \\ \mathbf{u}^*/2 \end{bmatrix} + \begin{bmatrix} \beta' \\ \gamma' \\ \mathbf{u}' \end{bmatrix}
\tag{18}
$$

where $\beta' := z_i^*$, $\gamma' := z_s^*$ and

$$
u_p' := \begin{cases} z_p^* & \text{if } p \in [k] \setminus \{i, s\} \\ 0 & \text{otherwise.} \end{cases}
$$

First we argue that $(\bar{\beta}, \bar{\gamma}, \bar{\mathbf{u}})$ is feasible for (Di-aux). From (18), we have

$$
\begin{aligned}
\bar{\beta}\boldsymbol{\theta}^i + \bar{\gamma}\mathbf{e_s} + \Theta^T\bar{\mathbf{u}} &= \beta^*\boldsymbol{\theta}^i + \Theta^T\mathbf{u}^*/2 + \beta'\boldsymbol{\theta}^i + \gamma'\mathbf{e_s} + \Theta^T\mathbf{u}' \\
&= \mathbf{c} - \mathbf{r} + \beta'\boldsymbol{\theta}^i + \gamma'\mathbf{e_s} + \Theta^T\mathbf{u}' && (\because (\beta^*, \mathbf{u}^*) \text{ is feasible for (Di)}) \\
&= \mathbf{c} - \mathbf{r} + \bar{I}_k^T\mathbf{z}^* && (\text{using the definition of } (\beta', \gamma', \mathbf{u}')) \\
&= \mathbf{c}. && (\text{using (16)})
\end{aligned}
$$

To argue about the nonnegativity of $(\bar{\beta}, \bar{\gamma}, \bar{\mathbf{u}})$, it suffices to argue that

1. $z_i^* + \beta^* \geq 0$

2. $\mathbf{z}^*([k] \setminus \{i\}) \geq \mathbf{0}$.

Note that our assumption on $\eta$ implies $\eta < \frac{1}{2\sqrt{2k}}\frac{c_{\min}}{c_{\max}}$ and therefore Lemmas A.2 and A.3 apply.

We have

$$
\begin{aligned}
z_i^* &= \bar{I}^{-T}(i,i)r_i + \sum_{p \in [k] \setminus \{i\}} \bar{I}^{-T}(i,p)r_p \\
&\geq 0 + \sum_{p \in [k] \setminus \{i\}} \bar{I}^{-T}(i,p)r_p && (\because \bar{I}^{-T}(i,i) \geq 0, r_i \geq 0) \\
&\geq -2\sqrt{2}\eta k\frac{c_{\max}}{2}. && (\text{using (17) and Lemma A.3})
\end{aligned}
\tag{19}
$$

Combining the lower bound on $z_i^*$ with the lower bound on $\beta^*$ in Lemma A.2 we get

$$
\begin{aligned}
z_i^* + \beta^* &\geq c_i - 3\sqrt{2}\eta k c_{\max} \\
&\geq c_{\min} - 3\sqrt{2}\eta k c_{\max} \\
&> 0.
\end{aligned}
$$

The last inequality above follows from our assumption on $\eta$. Indeed, we have

$$
\begin{aligned}
\eta &< \frac{1}{3k}\left(\frac{c_{\min}}{c_{\max}} - \frac{1}{2}\right) \\
&< \frac{1}{3\sqrt{2}k}\frac{c_{\min}}{c_{\max}}. \qquad\qquad\qquad \left(\because \frac{c_{\min}}{c_{\max}} \leq 1\right)
\end{aligned}
$$

Similarly, for any $t \in [k] \setminus \{i\}$ we have

$$
\begin{aligned}
z_t^* &\geq r_t - \|\bar{I}^{-T} - I\|_\infty \|\mathbf{r}\|_\infty && \text{(using (16))} \\
&\geq r_t - 2\sqrt{2}\eta k \frac{c_{\max}}{2} && \text{(using (17) and Lemma A.4)} \\
&\geq c_{\min} - \frac{\eta}{1-\eta}c_{\max} - 2\sqrt{2}\eta k \frac{c_{\max}}{2}. && \text{(using Lemma A.3)}
\end{aligned}
\qquad (20)
$$

Our assumption on $\eta$ yields a positive lower bound on the above expression. Indeed, we have

$$
\begin{aligned}
c_{\min} &> \frac{c_{\max}}{2} + 3k\eta c_{\max} && \text{(by assumption on } \eta\text{)} \\
&\geq \frac{c_{\max}}{2} + 2(k+1)\eta c_{\max} && (\because k \geq 2) \\
&= \frac{c_{\max}}{2} + 2\eta c_{\max} + 2\eta k c_{\max} \\
&\geq \frac{c_{\max}}{2} + \frac{\eta}{1-\eta}c_{\max} + \sqrt{2}\eta k c_{\max} && (\because \eta \leq 1/2)
\end{aligned}
$$

Using the above in (20), we get

$$
z_t^* > c_{\max}/2. \qquad (21)
$$

Therefore $(\bar{\beta}, \bar{\gamma}, \bar{\mathbf{u}})$ is feasible for (Di-aux).

Now we argue that the objective value of (Di-aux) at $(\bar{\beta}, \bar{\gamma}, \bar{\mathbf{u}})$ is greater than $\beta^*$. Indeed note that

$$
\begin{aligned}
\beta' + (2\sqrt{2}\eta k)\gamma' &= z_i^* + (2\sqrt{2}\eta k)z_s^* \\
&> -\sqrt{2}\eta k c_{\max} + 2\sqrt{2}\eta k \frac{c_{\max}}{2} && \text{(using (19) and (21))} \\
&= 0.
\end{aligned}
$$

That is, $\beta' + (2\sqrt{2}\eta k)\gamma' > 0$ or equivalently, $\bar{\beta} + (2\sqrt{2}\eta k)\bar{\gamma} > \beta^*$ thereby concluding the proof. $\square$

**Lemma A.6.** *Suppose* $\frac{c_{\min}}{c_{\max}} > \frac{1}{2}$ *and* $\eta < \frac{1}{4k}\left(\frac{c_{\min}}{c_{\max}} - \frac{1}{2}\right)$. *Then for any* $s \in [k] \setminus \{i\}$, *if* $y_s^*$ *is negative, we have*

$$
y_s^* > -4\sqrt{2}\eta k. \qquad (22)
$$

*Proof.* Pick any $s \in [k] \setminus \{i\}$ such that $y_s^* < 0$. Consider the auxiliary LP

$$
\begin{aligned}
\min \quad & \mathbf{c}^T \mathbf{y} \\
\text{s.t.} \quad & \Theta \mathbf{y} \geq \mathbf{0} \\
& \mathbf{y}^T \boldsymbol{\theta}^i \geq 1 \\
& y_s \leq -4\sqrt{2}\eta k
\end{aligned}
\qquad \text{(Pi-aux)}
$$

and its dual

$$
\begin{aligned}
\max \quad & \beta + (4\sqrt{2}\eta k)\gamma \\
\text{s.t.} \quad & \beta\boldsymbol{\theta}^i - \gamma\mathbf{e_s} + \Theta^T\mathbf{u} = \mathbf{c} \\
& \beta, \gamma, \mathbf{u} \geq 0.
\end{aligned}
\tag{Di-aux}
$$

If we show that $\mathbf{y}^*$ is not an optimal solution to (Pi-aux), then we can conclude that $y_s^* > -4\sqrt{2}\eta k$. Therefore our goal is to show that the optimal value of (Pi-aux) is greater than $\mathbf{c}^T\mathbf{y}^*$. Equivalently, we may also show that the optimal value of (Di-aux) is greater than $\beta^*$. We do so by constructing a feasible solution for (Di-aux) at which the objective value is greater than $\beta^*$.

Let $\mathbf{z}^*$ be the solution to the system

$$
I'^T\mathbf{z} = \mathbf{r} + \frac{c_{\max}}{2}\mathbf{e_s}
\tag{23}
$$

where recall that $\mathbf{r} = \Theta^T\mathbf{u}^*/2$.

Note that the rows of $I'$ belong to the unit simplex and for any $i \in [k]$, we have

$$
\begin{aligned}
\|I'(i,:) - \mathbf{e}^i\|_\infty &\leq \eta \\
&\leq \frac{1}{2\sqrt{2k}}. \quad \text{(by assumption on } \eta\text{)}
\end{aligned}
$$

Therefore using Lemma A.1, we conclude that

$$
\|I'^{-T} - I\|_\infty \leq 2\sqrt{2}\eta k.
\tag{24}
$$

Define the point

$$
\begin{bmatrix} \bar{\beta} \\ \bar{\gamma} \\ \bar{\mathbf{u}} \end{bmatrix} := \begin{bmatrix} \beta^* \\ 0 \\ \mathbf{u}^*/2 \end{bmatrix} + \begin{bmatrix} \beta' \\ c_{\max}/2 \\ \mathbf{u}' \end{bmatrix}
\tag{25}
$$

where $\beta' := z_i^*$ and

$$
u_p' := \begin{cases} z_p^* & \text{if } p \in [k] \setminus \{i\} \\ 0 & \text{otherwise.} \end{cases}
$$

First we argue that $(\bar{\beta}, \bar{\gamma}, \bar{\mathbf{u}})$ is feasible for (Di-aux). From (25), we have

$$
\begin{aligned}
\bar{\beta}\boldsymbol{\theta}^i - \bar{\gamma}\mathbf{e_s} + \Theta^T\bar{\mathbf{u}} &= \beta^*\boldsymbol{\theta}^i + \Theta^T\mathbf{u}^*/2 + \beta'\boldsymbol{\theta}^i - c_{\max}\mathbf{e_s}/2 + \Theta^T\mathbf{u}' && \\
&= \mathbf{c} - \mathbf{r} + \beta'\boldsymbol{\theta}^i - c_{\max}\mathbf{e_s}/2 + \Theta^T\mathbf{u}' && (\because (\beta^*, \mathbf{u}^*) \text{ is feasible for (Di))} \\
&= \mathbf{c} - \mathbf{r} + I'^T\mathbf{z}^* - c_{\max}\mathbf{e_s}/2 && \text{(using the definition of } (\beta', \mathbf{u}')) \\
&= \mathbf{c}. && \text{(using (23))}
\end{aligned}
$$

To argue about the nonnegativity of $(\bar{\beta}, \bar{\gamma}, \bar{\mathbf{u}})$, it suffices to argue that

1. $z_i^* + \beta^* \geq 0$

2. $\mathbf{z}^*([k] \setminus \{i\}) \geq \mathbf{0}$.

Note that our assumption on $\eta$ implies $\eta < \frac{1}{2\sqrt{2k}}\frac{c_{\min}}{c_{\max}}$ and therefore Lemmas A.2 and A.3 apply.

We have

$$
\begin{aligned}
z_i^* &= I'^{-T}(i,i)r_i + I'^{-T}(i,s)(r_s + c_{\max}/2) + \sum_{p\in[k]\setminus\{i,s\}} I'^{-T}(i,p)r_p && \\
&\geq 0 + I'^{-T}(i,s)(r_s + c_{\max}/2) + \sum_{p\in[k]\setminus\{i,s\}} I'^{-T}(i,p)r_p && (\because I'^{-T}(i,i) \geq 0, r_i \geq 0) \\
&\geq -2\sqrt{2}\eta k c_{\max}. && \text{(using (24) and Lemma A.3)}
\end{aligned}
\tag{26}
$$

Combining the lower bound on $z_i^*$ with the lower bound on $\beta^*$ in Lemma A.2 yields

$$
\begin{aligned}
z_i^* + \beta^* &\geq c_i - 4\sqrt{2}\eta k c_{\max} \\
&\geq c_{\min} - 4\sqrt{2}\eta k c_{\max} \\
&> 0.
\end{aligned}
$$

The last inequality above follows from our assumption on $\eta$. Indeed, we have

$$
\begin{aligned}
\eta &< \frac{1}{4k}\left(\frac{c_{\min}}{c_{\max}} - \frac{1}{2}\right) \\
&< \frac{1}{4\sqrt{2}k}\frac{c_{\min}}{c_{\max}}. \qquad\qquad \left(\because \frac{c_{\min}}{c_{\max}} \leq 1\right)
\end{aligned}
$$

Similarly, for any $t \in [k] \setminus \{i\}$ we have

$$
\begin{aligned}
z_t^* &\geq r_t + c_{\max} I(s,t)/2 - \|I'^{-T} - I\|_\infty \|\mathbf{r} + c_{\max}\mathbf{e_s}/2\|_\infty & \text{(using (23))} \\
&\geq r_t - \|I'^{-T} - I\|_\infty \|\mathbf{r} + c_{\max}\mathbf{e_s}/2\|_\infty & \\
&\geq r_t - 2\sqrt{2}\eta k c_{\max} & \text{(using (24) and Lemma A.4)} \quad (27) \\
&\geq c_{\min} - \frac{\eta}{1-\eta}c_{\max} - 2\sqrt{2}\eta k c_{\max}. & \text{(using Lemma A.3)}
\end{aligned}
$$

Our assumption on $\eta$ yields a positive lower bound on the above expression. Indeed, we have

$$
\begin{aligned}
c_{\min} &> \frac{c_{\max}}{2} + 4k\eta c_{\max} & \text{(by assumption on } \eta) \\
&\geq \frac{c_{\max}}{2} + (2+3k)\eta c_{\max} & (\because k \geq 2) \\
&= \frac{c_{\max}}{2} + 2\eta c_{\max} + 3\eta k c_{\max} & \\
&\geq \frac{c_{\max}}{2} + \frac{\eta}{1-\eta}c_{\max} + 2\sqrt{2}\eta k c_{\max} & (\because \eta \leq 1/2)
\end{aligned}
$$

Using the above in (27), we get

$$
z_t^* > c_{\max}/2. \qquad (28)
$$

Therefore $(\bar\beta, \bar\gamma, \bar{\mathbf{u}})$ is feasible for (Di-aux).

Now we argue that the objective value of (Di-aux) at $(\bar\beta, \bar\gamma, \bar{\mathbf{u}})$ is greater than $\beta^*$. Indeed note that

$$
\begin{aligned}
\beta' + (4\sqrt{2}\eta k)\frac{c_{\max}}{2} &= z_i^* + (4\sqrt{2}\eta k)\frac{c_{\max}}{2} \\
&> -2\sqrt{2}\eta k c_{\max} + (4\sqrt{2}\eta k)\frac{c_{\max}}{2} & \text{(using (26))} \\
&= 0.
\end{aligned}
$$

That is, $\beta' + (4\sqrt{2}\eta k)\frac{c_{\max}}{2} > 0$ or equivalently, $\bar\beta + (4\sqrt{2}\eta k)\bar\gamma > \beta^*$ thereby concluding the proof. $\qquad\square$

**Lemma A.7.** *Suppose $\frac{c_{\min}}{c_{\max}} > \frac{1}{2}$ and $\eta < \frac{1}{4k}\left(\frac{c_{\min}}{c_{\max}} - \frac{1}{2}\right)$. Then*

$$
\frac{1 - 4\sqrt{2}\eta^2 k}{\theta_{ii}} \leq y_i^* \leq \frac{1 + 4\sqrt{2}\eta^2 k}{\theta_{ii}}. \qquad (29)
$$

*Proof.* We note that the constraint $\mathbf{y}^T\boldsymbol{\theta}^i \geq 1$ in (Pi) is tight at optimality. Indeed otherwise one may scale the optimal solution so as to make that constraint tight and obtain a strictly smaller objective value, thereby contradicting optimality.

Then we have

$$
\begin{aligned}
1 &= \mathbf{y}^{*T} \boldsymbol{\theta}^i \\
&= y_i^* \theta_{ii} + \sum_{s \in [k] \setminus \{i\}} y_s^* \theta_{is}.
\end{aligned}
\tag{30}
$$

Moreover

$$
\begin{aligned}
\left| \sum_{s \in [k] \setminus \{i\}} y_s^* \theta_{is} \right| &\leq \|\mathbf{y}^*([k] \setminus \{i\})\|_\infty \|\boldsymbol{\theta}^i([k] \setminus \{i\})\|_1 && \text{(using Hölder's inequality)} \\
&\leq \eta \|\mathbf{y}^*([k] \setminus \{i\})\|_\infty && (\because \|\boldsymbol{\theta}^i([k] \setminus \{i\})\|_1 \leq \eta) \\
&\leq 4\sqrt{2}\eta^2 k. && \text{(using Lemmas A.5 and A.6)}
\end{aligned}
\tag{31}
$$

Using (31) in (30) yields the desired result. $\qquad\square$

*Proof of Theorem 3.4.* First note that (P) is both feasible and bounded below, which implies that it has an optimal solution. By assumption, there exists a $k \times k$ submatrix of $\Theta$ whose entrywise distance from $I$ is at most $\eta$; this implies that the spectral norm distance of such a submatrix from $I$ is at most $\eta k$ which is, by assumption, at most $(c_{\min}/c_{\max} - 1/2)/4$ which is at most one. This implies that the column rank of $\Theta$ is $k$. Therefore, using the fact that $B$ is full-rank, we conclude that the column range of $\Theta$ is equal to the range of $P$ and consequently the rank of $P$ is $k$. Therefore (P) may be rewritten as

$$
\begin{aligned}
\min \quad & \mathbf{c}^T \mathbf{y} \\
\text{s.t.} \quad & \Theta \mathbf{y} \geq \mathbf{0} \\
& \mathbf{y}^T \boldsymbol{\theta}^i \geq 1.
\end{aligned}
\tag{Py}
$$

Note that (Py) is both feasible and bounded below, which implies that it has an optimal solution. Since $\mathbf{x}^*$ is an optimal solution to (P), there exists an optimal solution to (Py), called $\mathbf{y}^*$, satisfying $\Theta \mathbf{y}^* = \mathbf{x}^*$. Using Lemmas A.5, A.6, and A.7, we conclude that

$$
\left\| \mathbf{y}^* - \frac{\mathbf{e}_j}{\theta_{ij}} \right\|_\infty \leq \sqrt{2}\eta k \max\{2, 4, 4\eta/\theta_{ij}\} = 4\sqrt{2}\eta k.
\tag{32}
$$

The last equality above holds because $\theta_{ij} \geq 1 - \eta$ and $\eta < 1/2$. Then we have

$$
\begin{aligned}
\left\| \mathbf{x}^* - \frac{\boldsymbol{\theta}_j}{\theta_{ij}} \right\|_\infty &= \left\| \Theta \mathbf{y}^* - \Theta \frac{\mathbf{e}_j}{\theta_{ij}} \right\|_\infty \\
&\leq \|\Theta\|_\infty \left\| \mathbf{y}^* - \frac{\mathbf{e}_j}{\theta_{ij}} \right\|_\infty \\
&\leq 4\sqrt{2}\eta k. && (\|\Theta\|_\infty = 1 \text{ and using (32))}
\end{aligned}
\tag{33}
$$

Lastly, we have

$$\left\| \frac{\mathbf{x}^*}{\|\mathbf{x}^*\|_\infty} - \boldsymbol{\theta}_j \right\|_\infty \le \left\| \frac{\mathbf{x}^*}{\|\mathbf{x}^*\|_\infty} - \mathbf{x}^* \right\|_\infty + \left\| \mathbf{x}^* - \frac{\boldsymbol{\theta}_j}{\theta_{ij}} \right\|_\infty + \left\| \frac{\boldsymbol{\theta}_j}{\theta_{ij}} - \boldsymbol{\theta}_j \right\|_\infty$$

(using triangle inequality)

$$= |1 - \|\mathbf{x}^*\|_\infty| + \left\| \mathbf{x}^* - \frac{\boldsymbol{\theta}_j}{\theta_{ij}} \right\|_\infty + \left\| \frac{\boldsymbol{\theta}_j}{\theta_{ij}} - \boldsymbol{\theta}_j \right\|_\infty$$

$$\le \left| 1 - \frac{\|\boldsymbol{\theta}_j\|_\infty}{\theta_{ij}} \right| + \left| \|\mathbf{x}^*\|_\infty - \frac{\|\boldsymbol{\theta}_j\|_\infty}{\theta_{ij}} \right| + \left\| \mathbf{x}^* - \frac{\boldsymbol{\theta}_j}{\theta_{ij}} \right\|_\infty + \left\| \frac{\boldsymbol{\theta}_j}{\theta_{ij}} - \boldsymbol{\theta}_j \right\|_\infty$$

(using triangle inequality)

$$\le \left| 1 - \frac{\|\boldsymbol{\theta}_j\|_\infty}{\theta_{ij}} \right| + 2 \left\| \mathbf{x}^* - \frac{\boldsymbol{\theta}_j}{\theta_{ij}} \right\|_\infty + \left\| \frac{\boldsymbol{\theta}_j}{\theta_{ij}} - \boldsymbol{\theta}_j \right\|_\infty$$

(using reverse triangle inequality)

$$\le \left( \frac{\|\boldsymbol{\theta}_j\|_\infty}{\theta_{ij}} - 1 \right) + 8\sqrt{2}\eta k + \left( \frac{1}{\theta_{ij}} - 1 \right) \|\boldsymbol{\theta}_j\|_\infty$$

$$\le 8\sqrt{2}\eta k + 2 \left( \frac{1}{\theta_{ij}} - 1 \right)$$

$$\le 8\sqrt{2}\eta k + \frac{2\eta}{1 - \eta}$$

$$< 8\sqrt{2}\eta k + 4\eta$$

$$= 4\eta(2\sqrt{2}k + 1)$$

where the inequality in the fifth line from bottom follows from using (33), the inequality in the fourth line from bottom follows because $\|\boldsymbol{\theta}_j\|_\infty \le 1$, the inequality in the third line from bottom follows because $\theta_{ij} \ge 1 - \eta$, and the inequality in the second line from bottom follows because $\eta < 1/2$.

Lastly we provide an argument for the time complexity claim. Since the rank of $P$ is $k$, the column range of $P$ is same as the column range of $V$ where $V$ is an $n \times k$ matrix whose columns contain the eigenvectors of $P$ corresponding to its $k$ nonzero eigenvalues. This implies that (P) is equivalent to $\{\min \mathbf{e}^T (V\mathbf{y}) \text{ subject to } V\mathbf{y} \ge 0, (V\mathbf{y})_{\mathcal{J}(i)} \ge 1\}$ which contains $n + 1$ constraints and $k$ variables. Hence the result in Megiddo [1984] implies that (P) can be solved in $\mathcal{O}(n)$ time. Moreover, $V$ can be obtained from $P$ in $\mathcal{O}(n^2)$ time using, for instance, randomized SVD techniques (Halko et al. [2011]). $\qquad\square$

## B Some Concentration Properties in the MMSB

In this section, we show concentration properties of some key random variables associated with random matrices $\Theta$ and $\Theta B$. We shall use these observations for our subsequent proofs, but they may also be of independent interest. Even though we work the equal parameter Dirichlet distribution, the proof techniques here easily extend to the case with different Dirichlet parameters.

Define $l := \sigma_{\min}(B)$ and $u := \sigma_{\max}(B)$. Suppose the $k$ parameters of the Dirichlet distribution are all equal to $\alpha$. We repeatedly use the facts that for any $i \in [n]$, $s \in [k]$,

$$\mathbb{E}[\theta_{is}] = \frac{1}{k} \tag{34}$$

and

$$\mathbb{E}[\theta_{is}^2] = \frac{\alpha + 1}{k(\alpha k + 1)}. \tag{35}$$

Moreover, if $s, t \in [k]$ such that $s \neq t$ then

$$\mathbb{E}[\theta_{is}\theta_{it}] = \frac{\alpha}{k(\alpha k + 1)}. \tag{36}$$

**Lemma B.1.** *For any* $j \in [k]$, *we have* $\frac{9}{10}\frac{n}{k} \leq c_j \leq \frac{11}{10}\frac{n}{k}$ *with probability at least* $1 - 2\exp\left(\frac{-n}{50k^2}\right)$.

*Proof.* For any $j \in [k]$, $c_j$ is the sum of $n$ independent bounded random variables $\{\theta_{ij}\}_{i=1}^n$. Indeed each row of $\Theta$ is sampled independently and each entry of $\Theta$ lies in $[0, 1]$. Moreover, using (34) we get that $\mathbb{E}[c_j] = n/k$. Thus, using Hoeffding's inequality, we have that for any $z > 0$

$$\Pr(|c_j - n/k| \geq z) \leq 2\exp\left(\frac{-2z^2}{n}\right). \tag{37}$$

Setting $z = n/10k$ in (37) yields the desired result. $\square$

**Corollary B.2.** *We have* $c_{\min}/c_{\max} \geq 9/11$ *with probability at least* $1 - p_1$, *where* $p_1 := 2k\exp\left(\frac{-n}{50k^2}\right)$.

*Proof.* Lemma B.1 implies that with probability at least $1 - 2k\exp\left(\frac{-n}{50k^2}\right)$, both $c_{\min} \geq 9n/10k$ and $c_{\max} \leq 11n/10k$ hold. $\square$

**Lemma B.3.** *For any* $\epsilon > 0$, $\|\Theta B\| \leq u\sqrt{\frac{2n}{k}} + \epsilon\|\Theta\|$ *with probability at least* $1 - \left(\frac{2u}{\epsilon} + 1\right)^k \exp\left(\frac{-2n}{k^2}\right)$.

For proving Lemma B.3, we first prove the following statements for set $\mathcal{C} := \{\mathbf{y} \in \mathbb{R}^k : \exists\, \mathbf{x} \in \mathbb{R}^k$ such that $B\mathbf{x} = \mathbf{y}, \|\mathbf{x}\| = 1\}$ defined as the image of the unit sphere under $B$.

**Lemma B.4.** *If* $\mathcal{E}$ *is an* $\epsilon$-*net of* $\mathcal{C}$ *of smallest possible cardinality, then* $|\mathcal{E}| \leq \left(\frac{2u}{\epsilon} + 1\right)^k$.

*Proof.* Let $\mathcal{E}'$ be a maximal $\epsilon$-separated subset of $\mathcal{C}$. Note that by definition of an $\epsilon$-separated subset, for any distinct $\mathbf{x}, \mathbf{y} \in \mathcal{E}'$, we have $\|\mathbf{x} - \mathbf{y}\| > \epsilon$. Moreover, the maximality of $\mathcal{E}'$ implies that $\mathcal{E}'$ is also an $\epsilon$-net of $\mathcal{C}$. Therefore

$$|\mathcal{E}| \leq |\mathcal{E}'|. \tag{38}$$

We also have that the union of $|\mathcal{E}'|$ disjoint balls $\bigcup_{\mathbf{x} \in \mathcal{E}'} \mathcal{B}(\mathbf{x}, \epsilon/2) \subseteq \mathcal{C} + \mathcal{B}(\mathbf{0}, \epsilon/2) \subseteq \mathcal{B}(\mathbf{0}, u + \epsilon/2)$. Therefore

$$\text{vol}\left(\bigcup_{\mathbf{x} \in \mathcal{E}'} \mathcal{B}(\mathbf{x}, \epsilon/2)\right) \leq \text{vol}(\mathcal{B}(\mathbf{0}, u + \epsilon/2)) \tag{39}$$

which implies that $|\mathcal{E}'|(\epsilon/2)^k \leq (u + \epsilon/2)^k$ which yields the desired result when combined with (38). $\square$

**Lemma B.5.** *Suppose* $\mathbf{y} \in \mathcal{C}$. *For any* $i \in [n]$:

1. $0 \leq \langle \boldsymbol{\theta}^i, \mathbf{y}\rangle^2 \leq u^2$

2. $\frac{l^2}{k(\alpha k + 1)} \leq \mathbb{E}[\langle \boldsymbol{\theta}^i, \mathbf{y}\rangle^2] \leq \frac{u^2}{k}$

*Proof.* Let $\mathbf{y} = B\mathbf{x}$ such that $\|\mathbf{x}\| = 1$. Then $l \leq \|\mathbf{y}\| \leq u$.

1. We have

$$\begin{aligned}
\langle \boldsymbol{\theta}^i, \mathbf{y}\rangle^2 &\leq \|\boldsymbol{\theta}_i\|^2 \|\mathbf{y}\|^2 && \text{(using Cauchy-Schwarz inequality)}\\
&\leq u^2 && (\|\boldsymbol{\theta}_i\| \leq 1).
\end{aligned}$$

2. We have

$$\mathbb{E}[\langle \boldsymbol{\theta}^i, \mathbf{y} \rangle^2] = \mathbb{E}[\theta_{i1}^2 y_1^2 + \cdots + \theta_{ik}^2 y_k^2] + \mathbb{E}\left[ \sum_{\substack{s,t \in [k]: \\ s \neq t}} \theta_{is}\theta_{it}y_s y_t \right]$$

$$= \frac{\alpha+1}{k(\alpha k+1)} \|\mathbf{y}\|^2 + \mathbb{E}\left[ \sum_{\substack{s,t \in [k]: \\ s \neq t}} \theta_{is}\theta_{it}y_s y_t \right] \qquad \text{(using (35))}$$

$$= \frac{\alpha+1}{k(\alpha k+1)} \|\mathbf{y}\|^2 + \frac{\alpha}{k(\alpha k+1)} \sum_{\substack{s,t \in [k]: \\ s \neq t}} y_s y_t \qquad \text{(using (36))}$$

$$= \frac{1}{k(\alpha k+1)} \|\mathbf{y}\|^2 + \frac{\alpha}{k(\alpha k+1)} (\mathbf{e}^T \mathbf{y})^2 \qquad \text{(re-arranging terms)}.$$

Now noting the second term on the right hand side above is nonnegative yields the desired lower bound.

Similarly noting that $\mathbf{e}^T \mathbf{y} \leq u\sqrt{k}$ (using Cauchy-Schwarz inequality) yields the desired upper bound.

$\square$

*Proof of Lemma B.3.* We have
$$\|\Theta B\| = \sup_{\mathbf{x} \in S^{k-1}} \|\Theta B\mathbf{x}\| = \sup_{\mathbf{y} \in \mathcal{C}} \|\Theta \mathbf{y}\|. \tag{40}$$
Let $\mathcal{E}$ denote an $\epsilon$-net of $\mathcal{C}$ of smallest possible cardinality. Then we have
$$\|\Theta B\| \leq \sup_{\mathbf{y} \in \mathcal{E}} \|\Theta \mathbf{y}\| + \epsilon\|\Theta\|. \tag{41}$$
Indeed if the supremum defining $\|\Theta B\|$ on the RHS in (40) is attained at $\mathbf{y}_s$, and if $\mathbf{y}_e$ is a point in $\mathcal{E}$ such that $\|\mathbf{y}_s - \mathbf{y}_e\| \leq \epsilon$, then
$$\begin{aligned} \|\Theta B\| &= \|\Theta \mathbf{y}_s\| \\ &= \|\Theta \mathbf{y}_e + \Theta(\mathbf{y}_s - \mathbf{y}_e)\| \\ &\leq \|\Theta \mathbf{y}_e\| + \|\Theta(\mathbf{y}_s - \mathbf{y}_e)\| \qquad \text{(using triangle inequality)} \\ &\leq \sup_{\mathbf{y} \in \mathcal{E}} \|\Theta \mathbf{y}\| + \epsilon\|\Theta\|. \end{aligned}$$
For any $\mathbf{y} \in \mathcal{E}$, we have
$$\|\Theta \mathbf{y}\|^2 = \langle \boldsymbol{\theta}^1, \mathbf{y} \rangle^2 + \cdots + \langle \boldsymbol{\theta}^n, \mathbf{y} \rangle^2.$$
Now note that $\|\Theta \mathbf{y}\|^2$ is the sum of $n$ independent random variables. Indeed using Lemma B.5 we conclude that each of these random variables is bounded and that $\mathbb{E}[\|\Theta \mathbf{y}\|^2] \leq \frac{nu^2}{k}$. Thus, using Hoeffding's inequality, we have that for any $z > 0$,
$$\Pr\left( \|\Theta \mathbf{y}\|^2 \geq \frac{nu^2}{k} + z \right) \leq \Pr(\|\Theta \mathbf{y}\|^2 \geq \mathbb{E}[\|\Theta \mathbf{y}\|^2] + z)$$
$$\leq \exp\left( \frac{-2z^2}{nu^4} \right).$$
Then using the union bound over the $\epsilon$-net, we obtain that
$$\Pr\left( \sup_{\mathbf{y} \in \mathcal{E}} \|\Theta \mathbf{y}\| \geq \sqrt{\frac{nu^2}{k} + z} \right) \leq |\mathcal{E}| \exp\left( \frac{-2z^2}{nu^4} \right)$$
$$\leq \left( \frac{2u}{\epsilon} + 1 \right)^k \exp\left( \frac{-2z^2}{nu^4} \right) \qquad \text{(using Lemma B.4)}$$

Setting $z = nu^2/k$ in the above, we note that $\sup_{\mathbf{y} \in \mathcal{E}} \|\Theta y\| \leq u\sqrt{\dfrac{2n}{k}}$ with probability at least $1 -$

$\left(\dfrac{2u}{\epsilon} + 1\right)^k \exp\left(\dfrac{-2n}{k^2}\right)$, combining which with (41) yields the desired result. $\qquad\square$

**Corollary B.6.** $\|\Theta\| \leq 2\sqrt{\dfrac{2n}{k}}$ with probability at least $1 - p_2$, where $p_2 := 5^k \exp\left(\dfrac{-2n}{k^2}\right)$.

*Proof.* Set $B = I$ and $\epsilon = 1/2$ in Lemma B.3. $\qquad\square$

**Corollary B.7.** $\|\Theta B\| \leq 2u\sqrt{\dfrac{2n}{k}}$ with probability at least $1 - p_2$.

*Proof.* This follows simply from using the inequality $\|\Theta B\| \leq \|\Theta\|\|B\|$ and the upper bound obtained in Corollary B.6. $\qquad\square$

**Lemma B.8.** $\sigma_k(\Theta B) \geq \dfrac{1}{4}\dfrac{l}{\sqrt{\alpha k + 1}}\sqrt{\dfrac{2n}{k}}$ with probability at least $1 - p_3$, where $p_3 := p_2 +$

$\left(\dfrac{16u\sqrt{\alpha k + 1}}{l} + 1\right)^k \exp\left(\dfrac{-nl^4}{2k^2 u^4(\alpha k + 1)^2}\right)$.

*Proof.* We have
$$\sigma_k(\Theta B) = \inf_{\mathbf{x} \in S^{k-1}} \|\Theta B\mathbf{x}\| = \inf_{\mathbf{y} \in \mathcal{C}} \|\Theta \mathbf{y}\|. \tag{42}$$
Let $\mathcal{E}$ denote an $\epsilon$-net of $\mathcal{C}$ of smallest possible cardinality. Then we have
$$\sigma_k(\Theta B) \geq \inf_{\mathbf{y} \in \mathcal{E}} \|\Theta \mathbf{y}\| - \epsilon\|\Theta\|. \tag{43}$$
Indeed if the infimum defining $\sigma_k(\Theta B)$ on the RHS in (42) is attained at $\mathbf{y}_s$, and if $\mathbf{y}_e$ is a point in $\mathcal{E}$ such that $\|\mathbf{y}_s - \mathbf{y}_e\| \leq \epsilon$, then
$$
\begin{aligned}
\sigma_k(\Theta B) &= \|\Theta \mathbf{y}_s\| \\
&= \|\Theta \mathbf{y}_e + \Theta(\mathbf{y}_s - \mathbf{y}_e)\| \\
&\geq |\|\Theta \mathbf{y}_e\| - \|\Theta(\mathbf{y}_s - \mathbf{y}_e)\|| \qquad \text{(using reverse triangle inequality)} \\
&\geq \|\Theta \mathbf{y}_e\| - \|\Theta(\mathbf{y}_s - \mathbf{y}_e)\| \\
&\geq \inf_{\mathbf{y} \in \mathcal{E}} \|\Theta \mathbf{y}\| - \epsilon\|\Theta\|.
\end{aligned}
$$
For any $\mathbf{y} \in \mathcal{E}$, we have
$$\|\Theta \mathbf{y}\|^2 = \langle \boldsymbol{\theta}^1, \mathbf{y}\rangle^2 + \cdots + \langle \boldsymbol{\theta}^n, \mathbf{y}\rangle^2.$$
Now note that $\|\Theta \mathbf{y}\|^2$ is the sum of $n$ independent bounded random variables. Indeed using Lemma B.5 we conclude that each of these random variables is bounded and that $\mathbb{E}[\|\Theta y\|^2] \geq \dfrac{nl^2}{k(\alpha k + 1)}$. Thus, using Hoeffding's inequality, we have that for any $z > 0$,
$$
\begin{aligned}
\Pr\left(\|\Theta y\|^2 \leq \dfrac{nl^2}{k(\alpha k + 1)} - z\right) &\leq \Pr(\|\Theta y\|^2 \leq \mathbb{E}[\|\Theta y\|^2] - z) \\
&\leq \exp\left(\dfrac{-2z^2}{nu^4}\right).
\end{aligned}
$$
Then using the union bound over the $\epsilon$-net, we obtain that
$$
\begin{aligned}
\Pr\left(\inf_{y \in \mathcal{E}} \|\Theta y\| \leq \sqrt{\dfrac{nl^2}{k(\alpha k + 1)}} - z\right) &\leq |\mathcal{E}| \exp\left(\dfrac{-2z^2}{nu^4}\right) \\
&\leq \left(\dfrac{2u}{\epsilon} + 1\right)^k \exp\left(\dfrac{-2z^2}{nu^4}\right) \qquad \text{(using Lemma B.4)}
\end{aligned}
$$

Setting $z = \dfrac{1}{2}\dfrac{nl^2}{k(\alpha k + 1)}$, we note that $\inf_{y \in \mathcal{E}} \|\Theta y\| \geq \sqrt{\dfrac{1}{2}\dfrac{nl^2}{k(\alpha k + 1)}}$ with probability at least

$1 - \left(\dfrac{2u}{\epsilon} + 1\right)^k \exp\left(\dfrac{-nl^4}{2k^2 u^4 (\alpha k + 1)^2}\right)$.

Using (43), we get that

$$\sigma_k(\Theta B) \geq \sqrt{\dfrac{1}{2}\dfrac{nl^2}{k(\alpha k + 1)}} - \epsilon\|\Theta\| \tag{44}$$

with probability at least $1 - \left(\dfrac{2u}{\epsilon} + 1\right)^k \exp\left(\dfrac{-nl^4}{2k^2 u^4 (\alpha k + 1)^2}\right)$.

Lastly, using the upper bound on $\|\Theta\|$ derived in Corollary B.6 in (44), we get that

$$\sigma_k(\Theta B) \geq \dfrac{1}{2}\dfrac{l}{\sqrt{\alpha k + 1}}\sqrt{\dfrac{2n}{k}} - 2\epsilon\sqrt{\dfrac{2n}{k}}$$

with probability at least $1 - p_2 - \left(\dfrac{2u}{\epsilon} + 1\right)^k \exp\left(\dfrac{-nl^4}{2k^2 u^4 (\alpha k + 1)^2}\right)$. Setting $\epsilon = \dfrac{1}{8}\dfrac{l}{\sqrt{\alpha k + 1}}$
yields the desired result. $\qquad\square$

## C  Proof of Main Theorem

In this section, we build the proof of Theorem 3.1.

**Lemma C.1.** *Let $p, \gamma \in (0, 1)$. If $n > \dfrac{\log(p/k)}{\log I_{1-\gamma}(\alpha, (k-1)\alpha)}$, then with probability at least $1 - p$, for each $j \in [k]$, there exists a row vector $\mathbf{r}^T$ in $\Theta$ such that*

$$\|\mathbf{r} - \mathbf{e}_j\|_\infty < \gamma. \tag{45}$$

*(Here $I_x(y, z)$ denotes the regularized incomplete beta function.)*

*Proof.* For any $j \in [k]$, define $E_j$ as the event that there exists a row $\mathbf{r}^T$ in $\Theta$ such that $\|\mathbf{r} - \mathbf{e}_j\|_\infty < \gamma$. Then for any $j \in [k]$, we have

$$
\begin{aligned}
\Pr(E_j^c) &= \prod_{i \in [n]} \Pr(\|\boldsymbol{\theta}^i - \mathbf{e}_j\|_\infty \geq \gamma) && (\because \text{rows of } \Theta \text{ are independently sampled})\\
&= \prod_{i \in [n]} \Pr(\theta_{ij} \leq 1 - \gamma) && (\because \text{rows of } \Theta \text{ belong to unit simplex})\\
&= [I_{1-\gamma}(\alpha, (k-1)\alpha)]^n && (I_x(y, z) \text{ is the CDF of marginal of Dirichlet distribution})\\
&< p/k. && (\text{by assumption on } n)
\end{aligned}
\tag{46}
$$

Therefore

$$
\begin{aligned}
\Pr(E_1 \cap \cdots \cap E_k) &= 1 - \Pr(E_1^c \cup \cdots \cup E_k^c)\\
&\geq 1 - \sum_{j \in [k]} \Pr(E_j^c) && (\text{using the union bound})\\
&> 1 - p. && (\text{using (46)})
\end{aligned}
$$

$\qquad\square$

*Proof of Theorem 3.1.* Using the lower bound assumption on $n$ and Lemma C.1, we conclude that with probability at least $1 - p$, for each $j \in [k]$, there exists a row $\mathbf{r}^T$ in $\Theta$ such that

$$\|\mathbf{r} - \mathbf{e}_j\|_\infty < \epsilon. \tag{47}$$

Recalling the definition of $\Delta$, we note that (47) is equivalent to

$$\|\Delta\|_{\max} < \epsilon. \tag{48}$$

Using Corollary B.7 and Lemma B.8, we conclude that

$$\kappa_0 \leq 8\kappa\sqrt{\alpha k + 1} \tag{49}$$

with probability at least $1 - p_2 - p_3$. Therefore (49) implies that

$$
\begin{aligned}
\min\left(\frac{1}{\sqrt{k-1}}, \frac{1}{2}\right) \frac{1}{2\sqrt{2}\kappa_0(1 + 80\kappa_0^2)} &\geq \epsilon_1 && \text{(using the definition of } \epsilon_1) \\
&> \epsilon && \text{(using the assumption on } \epsilon) \\
&> \|\Delta\|_{\max} && \text{(using (48))}
\end{aligned}
\tag{50}
$$

with probability at least $1 - p_2 - p_3$.

Using (50), we note that the assumption of Theorem 3.3 is satisfied with probability at least $1 - p - p_2 - p_3$. Therefore the set $\mathcal{J}$ returned by Algorithm 2 satisfies

$$
\begin{aligned}
\|\Pi\Theta(\mathcal{J},:) - I\|_{\max} &\leq 40\sqrt{2}\kappa_0^2\|\Delta\|_{\max} \\
&< 40\sqrt{2}\kappa_0^2\epsilon
\end{aligned}
\tag{51}
$$

with probability at least $1 - p - p_2 - p_3$ for some $k \times k$ permutation matrix $\Pi$.

Now from Corollary B.2, we know that

$$\frac{1}{4k}\left(\frac{c_{\min}}{c_{\max}} - \frac{1}{2}\right) \geq \frac{7}{88k} \tag{52}$$

with probability at least $1 - p_1$.

Thus we have

$$
\begin{aligned}
40\sqrt{2}\kappa_0^2\epsilon &< 40\sqrt{2}\kappa_0^2\epsilon_2 && \text{(using the assumption on } \epsilon) \\
&\leq 40\sqrt{2} \cdot 64\kappa^2(\alpha k + 1)\epsilon_2 && \text{(using (49))} \\
&= \frac{7}{88k} && \text{(using the definition of } \epsilon_2) \\
&\leq \frac{1}{4k}\left(\frac{c_{\min}}{c_{\max}} - \frac{1}{2}\right) && \text{(using (52))}
\end{aligned}
\tag{53}
$$

with probability at least $1 - p - p_1 - p_2 - p_3$. Combining (51) and (53), we conclude that the assumption of Theorem 3.4 is satisfied with probability at least $1 - p - p_1 - p_2 - p_3$. Therefore for any $j \in [k]$, the vector $\hat{\boldsymbol{\theta}}_j$ returned by SP+LP satisfies

$$
\begin{aligned}
\|\hat{\boldsymbol{\theta}}_j - \boldsymbol{\theta}_j\|_\infty &\leq 4 \cdot 40\sqrt{2}\kappa_0^2\epsilon \cdot (2\sqrt{2}k + 1) \\
&\leq 10240\sqrt{2}\kappa^2(\alpha k + 1)(2\sqrt{2}k + 1)\epsilon && \text{(using (49))} \\
&= \mathcal{O}(\alpha k^2\kappa^2\epsilon)
\end{aligned}
$$

with probability at least $1 - p - p_1 - p_2 - p_3$. Substituting the expressions for $p_1, p_2$ and $p_3$, the probability $1 - p - p_1 - p_2 - p_3$ can be expressed as $1 - p - c_1 e^{-c_2 n}$ such that $c_1, c_2$ are constants that depend on $\alpha, k, \kappa$.

$\square$

*Proof of Corollary 3.2.* From Theorem 3.1, we know that the maximum distance between vectors $\hat{\boldsymbol{\theta}}_1, \ldots, \hat{\boldsymbol{\theta}}_k$ and the columns of $\Theta$, up to a permutation, is $\mathcal{O}(\alpha k^2\kappa^2\epsilon)$ with probability at least $1 - p - c_1 e^{-c_2 n}$ where $c_1, c_2$ are constants that depend on $\alpha, k, \kappa$.

Similarly, the maximum distance between vectors $\hat{\boldsymbol{\theta}}_1, \ldots, \hat{\boldsymbol{\theta}}_k$ and the columns of $\bar{\Theta}$, up to a permutation, is $\mathcal{O}(\bar{\alpha}k^2\bar{\kappa}^2\bar{\epsilon})$ with probability at least $1 - \bar{p} - \bar{c}_1 e^{-\bar{c}_2 n}$ where $\bar{c}_1, \bar{c}_2$ are constants that depend on $\bar{\alpha}, k, \bar{\kappa}$.

Combining the above two observations with the triangle inequality and the union bound yields the desired result. $\square$