[Reviews · NeurIPS 2020]

Review 1

Summary and Contributions: In this paper the authors consider the overlapping community detection problem under the Mixed Membership Stochastic Block (MMSB) model. The input to the problem is a weighted matrix with weights resambing probabilities of interaction between nodes, and the number k of the communities to be found. It is assumed that there exists a ground truth community interaction k times k dimensional matrix B.The objective is to compute a partial assignment of each node to each other communities that best describes the input graph. Similarly to the Stochastic Block Model being the main underlying assumption when studying non-overlapping community detection with theoretical guarantees, its generalization, namely the MMSB model, is a reasonable model for studying overlapping community detection with theoretical guarantees. The main differentiation to prior work, is that the authors do not require the assumption that in each community there exists at least one node that belongs to no other community, which are called pure nodes. This assumption is arguably unrealistic in practice. It is known that if not pure nodes are present in the instance, then one cannot recover the community structures. The authors circumvent this barrier by showing that if two different sets of parameters to the MMSB that lead to the same probability matrix P, then the corresponding node-community fractional assignment are close with high probability. This means that the authors recover a node-community assignment that is close to the ground truth. The algorithm suggested by the authors provides some bounds on each node-community assignment.  The technical part is well-written, but it seems mostly accessible to people with a good knowledge of the techniques and tools that are used. The experimental part contains a well-executed set of experiments against a competitor algorithm (GeoNMF) which has been reported to outperform several other algorithms for recovery of the MMSB under the pure nodes assumption. The experimental setup contains several synthetic and real-world datasets of moderate size. The authors evaluate the tested algorithms as follows. They first generate an MMSB graph (according to some underlying node-community assignment) and then compare the produced solution to that of the ground truth data that generated the input. The results of the experiments highlight that the newly proposed algorithm performs better that the GeoNMF algorithm in terms of quality, but it suffers in efficiency. An interesting aspect of the paper is that they apply their algorithm on  a problem in computational biology, where one is given as an input a weighted graph with protein-protein interactions and the objective is to identify the overlapping communities of proteins that form what is called the protein complexes. This problem is well studied in computational biology and there are specialized metrics for evaluating the quality of the solution. The authors show that their algorithm performs comparably to some well-performing specialized algorithm for the problem, when evaluated on the specialized metrics for the problem. Unfortunately, the code used for the experiments is not available.

Strengths: First study to remove the assumption of pure nodes in each community for the problem, and provide provable guarantees. Experimental case-study on clustering proteins based on protein-protein interactions, showcases real-world applications of the proposed method.

Weaknesses: The scalability of the proposed method suffers compared to competitor algorithms. While the paper is certainly readable, it requires good knowledge of related tools and concepts. The probable guarantees provided in the paper only apply to inputs that are weighted graphs that follow the MMSB models, and for small values of k (the number of communities).

Correctness: I don't see any flaws in the claims. Unfortunately, I didn't manage to go through all the proofs in the appendix.

Clarity: In general, yes. A few parts could be improved. I explain in the comments below.

Relation to Prior Work: Yes, and in fact they explain well what differentiates the paper compared to the previous contributions.

Reproducibility: Yes

Additional Feedback: In Theorem 3.1, is it possible to state bounds where you do not hide O(1/poly k) factors inside \epsilon? In particular, is it easy to state the bound only with respect to k and \kappa? Could you provide a reference for the statement on line 119: "It is known that having pure nodes for each community is both necessary and sufficient for identifiability of MMSB" How do you perform the matrix-multiplication at step 5 of algorithm 2 in O(n^2) time? Could you elaborate more on how mild is the assumption that c_min / c_max > 1/2 in Theorem 3.4 is?


Review 2

Summary and Contributions: The proposed work studied the problem of obtaining node-community distribution matrix for Mixed Membership Stochastic Blockmodel (MMSB) to detect potentially overlapping network communities, without assuming the existence of pure nodes. The work proposed an SP+LP algorithm as the recovery procedure, which was supported with successful evaluation on both synthetic datasets and a real application. ====================== I have read the authors' response. I choose to keep my score.

Strengths: 1. The work proposed a theoretically interesting problem to study, i.e., how to detect potentially overlapping communities without assuming the existence of pure nodes for MMSB. 2. The work designed an algorithm to recover the node-community distribution matrix with theoretical support under certain assumptions. 3. The work demonstrated a successful application with the proposed model and algorithm.

Weaknesses: 1. Different community detection models have been proposed in literature other than MMSB. In practical applications, it is often preferred to see a comparison of effectiveness among these different models. As an example, Newman's modularity model and its subsequent extension on overlapping communities have been long used to detect network communities. Accordingly, could the modularity model be applied on the computational biology problem studied in this paper? 2. The claim of the LP's complexity (Line 134-135) in the proposed algorithm seems problematic to me. The "LP in linear time" claim made in Megiddo [1984] assumed that the variable dimension is fixed. While for the proposed SP+LP algorithm, the dimension is $n$, which seems inappropriate to be treated as a constant number. Let me know if there is misunderstanding here.

Correctness: The work seems technical sound to me, except the analysis of the proposed algorithm's time complexity (Ref. point 2 in "Weaknesses" section).

Clarity: Good.

Relation to Prior Work: Good.

Reproducibility: Yes

Additional Feedback: For questions to authors, please ref. "Weaknesses" section.


Review 3

Summary and Contributions: The paper proposes a new provable overlapping community detection algorithm based on the idea of MMSB, but without the assumption of pure nodes. The algorithm starts with a successive projection which selects almost pure node. Then, using that set of nodes the final estimation is generated through linear programming. The paper also proves the theoretical guarantees of the algorithm.

Strengths: -The idea is simple and it can find communities in the real datasets applied in the paper. -The major contribution is the theoretical guarantees of this algorithm, showing the converge to a correct solution.

Weaknesses: -One of the main problems is the poor performance of the algorithm against state of the art methods. Even though authors try to justify it, saying that ground truth in biology networks as not been achieved, then look for another type of network where the ground truth is known. -The algorithm is quite expensive, has the paper shows the time complexity is O(n^2), which is impossible to apply to medium-size networks. This is corroborated by the real experiment where the largest network has only 5000 nodes (the synthetic network). -The experiment section must be largely improved. Most results are not even discussed. Please, reduce the size of the graph (fit three plots in a row) and the setup, use this space to discuss real results. Also, add SVI, Bayesian variant of SNMF, OCCAM, and SAAC to the results. Even though these algorithms were beaten by GeoNMF, algorithms behave differently on datasets.

Correctness: Yes, they seem correct.

Clarity: -The main idea of the paper is clear, but algorithms not. Algorithms are not explained, not even cited (algorithm 1). You could save some space from the introduction and Problem formulation and expand this part.

Relation to Prior Work: Yes, it is clearly discussed.

Reproducibility: Yes

Additional Feedback: Please, read the following papers: An Analysis of Overlapping Community Detection Algorithms in Social Networks Community-Affiliation Graph Model for Overlapping Network Community Detection Overlapping Community Detection by Local Community Expansion, algorithm with time complexity O(m), where m is the number of edges. Also, this is a website with multiple datasets https://snap.stanford.edu/data/#onlinecoms

[Author Response · NeurIPS 2020]

We thank all three reviewers for their insightful feedback! We appreciate the positive comments regarding our main theoretical results (R2, R3, R4) and also our application case-study (R2, R3). We have addressed all reviewer comments in the following and will also incorporate the feedback in our paper. **@R2, restating bound of Theorem 3.1**: Yes. For instance, setting $\epsilon = \min\{\epsilon_1, \epsilon_2\}$, the bound can be further upper-bounded by $(1 + 2\sqrt{2}k)/(8\sqrt{k-1})$ exactly, provided $k \geq 5$. However, we think that doing so does not demonstrate the full potential of that theorem. In particular, we interpret $\epsilon$ as the desired recovery error level, and therefore not having $\epsilon$ in the RHS limits the error bound only in terms of problem parameter $k$ as already noted by the reviewer. **@R2, reference for the statement on line 119**: Please see Theorem 2.2 in Mao et al. [2020]. We shall also include this reference in our submitted paper. **@R2, matrix multiplication in step 5 of algorithm 2 in $O(n^2)$ time**: The matrix product in step 5 is $R - \frac{\mathbf{p}_{s'}\mathbf{p}_{s'}^T R}{\|\mathbf{p}_{s'}\|^2}$. Computing $\mathbf{p}_{s'}\mathbf{p}_{s'}^T R$ takes $O(n^2)$ time provided we first compute the matrix-vector product $\mathbf{p}_{s'}^T R$ and then right-multiply the resulting vector to $\mathbf{p}_{s'}$ (which is a vector outer product). Then computing $\|\mathbf{p}_{s'}\|^2$ and $\frac{\mathbf{p}_{s'}\mathbf{p}_{s'}^T R}{\|\mathbf{p}_{s'}\|^2}$ take $O(n)$ and $O(n^2)$ time respectively. Lastly, subtracting the matrix obtained in the previous sentence from $R$ also takes $O(n^2)$ time since we perform exactly $n^2$ subtractions. **@R2, mildness of assumption $c_{\min}/c_{\max} > 1/2$ in Theorem 3.4**: Corollary B.2 in the supplemental shows that this assumption is satisfied by a vast margin with very high probability when all $k$ Dirichlet parameters are equal. In fact, an analogous result can also be shown using the same techniques when some Dirichlet parameters are not equal. We highlight that this is an improvement to the recovery guarantee of competing GeoNMF which works only in the equal Dirichlet parameter setting. **@R2, broader impact**: We shall add more discussion on the broader impact. **@R3, comparison with modularity models for protein complex detection**: Indeed modularity-based methods may potentially be applied to protein complex detection. However, these methods are mainly heuristics and they suffer from the lack of recovery guarantees. Our main objective in this work is to advance a general-purpose mathematical theory for overlapping community detection. **@R3, LP in linear time claim (line 134-135)**: The LP effectively requires $\mathbf{x}$ to be in the column range of $P$. However, since $P = \Theta B \Theta^T$ (according to MMSB), it has rank $k$ with high probability. Consequently, the column range of $P$ is same as the column range of $Q$ where $Q$ is an $n \times k$ matrix whose columns denote the $k$ eigenvectors of $P$ corresponding to its nonzero eigenvalues. This implies that the LP in step 3 of algorithm 1 is equivalent to $\{\min \mathbf{e}^T(Q\mathbf{y}) \text{ subject to } Q\mathbf{y} \geq 0, (Q\mathbf{y})_{\mathcal{J}(i)} \geq 1\}$ which contains $n + 1$ constraints and whose dimension is $k$. Hence the Megiddo result implies that such LP can be solved in $O(n)$ time. Moreover, computing the $k$ largest eigenvectors is a step that appears in many algorithms in literature (including the competing GeoNMF) and can be achieved in $O(n^2)$ time using randomized SVD techniques (Halko et al. [2011]). We shall include these clarifying remarks and references in even more detail in our submitted paper. **@R4, using different real-world network, and poor performance against state-of-the-art algorithms**: To our best knowledge, there does not exist a real-world dataset of weighted graphs for which an independently derived ground-truth validation set also exists. If the R4 can point us to one, we would be very happy to use it in our future works. Regarding the performance against other methods, as mentioned in the paper, protein complex detection is a well-studied problem in computational biology for which many tailor-made heuristics exist. However, these heuristics do not come with any theoretical guarantees and as such the focus of our work is to advance a general-purpose mathematical theory which is not restricted to only protein complex detection. It may well be possible to tweak our algorithm so that better results for protein complex detection are obtained, but that is not the scope of this work. **@R4, time complexity $O(n^2)$ of presented algorithm is prohibitive, and largest synthetic network has only 5000 nodes**: We would like to clarify some factual misunderstandings: our largest synthetic network has 12000 nodes, and also, to our best knowledge, there does not exist a competing algorithm whose time complexity is under $O(n^2)$. We shall update our paper so that these aspects are more clear. Indeed in practice our algorithm runs slower compared to competing GeoNMF, and we mention improving this as a possible future direction. This increase in running time is traded-off with improved recovery guarantees; our recovery algorithm is effectively parameter-free, does not require pure nodes, allows for different community sizes and any general full-rank community interaction matrix $B$, and provides smaller recovery error. **@R4, adding more experiments since algorithms behave differently on different datasets, and more discussion of algorithm and results**: We shall update the paper as suggested to include a discussion of the algorithm and the results! While we agree that algorithms sometimes perform differently on different datasets, analyzing such patterns entails a much more comprehensive experimental comparison across diverse datasets and algorithms. Such a study is beyond the scope of this work since our main contribution is on the theoretical side. Therefore we compare our algorithm with a highly-cited domain-specific algorithm (ClusterONE). However this is a useful future work direction.

# References

Xueyu Mao, Purnamrita Sarkar, and Deepayan Chakrabarti. Estimating mixed memberships with sharp eigenvector deviations. *Journal of the American Statistical Association*, 0(just-accepted):1–24, 2020.

Nathan Halko, Per-Gunnar Martinsson, and Joel A Tropp. Finding structure with randomness: Probabilistic algorithms for constructing approximate matrix decompositions. *SIAM review*, 53(2):217–288, 2011.


[Meta-Review · NeurIPS 2020]

The authors introduce new interesting theoretical results for the Mixed Membership Stochastic Block (MMSB) model. The main differentiation to prior work, is that the authors drop the unrealistic assumption that in each community there exists at least one node belonging only to it. The main drawbacks of the paper is its applicability to real world scenario and its limited experimental analysis.